

**Measurement report: Atmospheric nitrate radical chemistry in the**
**South China Sea influenced by the urban outflow of the Pearl River**
**Delta**
Jie Wang[1, 2], Haichao Wang[1, 2 *], Yee Jun Tham[3, 4 *], Lili Ming[5], Zelong Zheng[1], Guizhen
Fang[3], Cuizhi Sun[1, 2], Zhenhao Ling[1, 2], Jun Zhao[1, 2], Shaojia Fan[1, 2]
[1] School of Atmospheric Sciences, Sun Yat-sen University, and Southern Marine Science
and Engineering Guangdong Laboratory (Zhuhai), Zhuhai, 519082, China
[2] Guangdong Provincial Observation and Research Station for Climate Environment and
Air Quality Change in the Pearl River Estuary, Key Laboratory of Tropical Atmosphere-
Ocean System (Sun Yat-sen University), Ministry of Education, Zhuhai, 519082, China
[3] School of Marine Sciences, Sun Yat-sen University, Zhuhai 519082, China.
[4] Pearl River Estuary Marine Ecosystem Research Station, Ministry of Education, Zhuhai,
519082, China.
[5] Technical Center of Gongbei Customs District of China, Zhuhai, 519000, China.
*Correspondence to:* Haichao Wang (wanghch27@mail.sysu.edu.cn); Yee Jun Tham
(thamyj@mail.sysu.edu.cn)
**Abstract.** Nitrate radical ($NO_3$) is a critical nocturnal atmospheric oxidant in the
troposphere, which widely affects the fate of air pollutants and regulates air quality. Many
previous works have reported the chemistry of $NO_3$ in inland regions of China, while less
study targets marine regions. Here, we present a field measurement of the $NO_3$ reservoir,
dinitrogen pentoxide ($N_2O_5$), and related species at a typical marine site (Da Wan Shan
Island) located in the South China Sea in the winter of 2021. Two patterns of air masses
were captured during the campaign, including the dominant airmass from inland China
(IAM) with a percentage of ~84%, and the airmass from eastern coastal areas (CAM) with
~16%. During the IAM period, the $NO_3$ production rate reached $1.6 \pm 0.9$ ppbv h$^{-1}$ due to
the transportation of the polluted urban plume with high $NO_x$ and $O_3$. While the average
nocturnal $N_2O_5$ and the calculated $NO_3$ mixing ratio were $119.5 \pm 128.6$ pptv and $9.9 \pm$
$12.5$ pptv, respectively, and the steady state lifetime of $NO_3$ was $0.5 \pm 0.7$ min on average,
indicating intensive nighttime chemistry and rapid $NO_3$ loss at this site. By examining the
reaction of $NO_3$ with volatile organic compounds (VOCs) and $N_2O_5$ heterogeneous
hydrolysis, we revealed that these two reaction pathways were not responsible for the $NO_3$
loss (<20%), since the $NO_3$ reactivity (k($NO_3$)) towards VOCs was small ($5.2\times10^{-3}$ s$^{-1}$)



and the aerosol loading was low. Instead, NO was proposed to significantly contribute to
nocturnal $NO_3$ loss at this site, despite the nocturnal NO concentration always at sub-ppbv
level and near the instrument detection limit. It might be from the local soil emission. We
infer that the nocturnal chemical $NO_3$ reactions would be largely enhanced once without
NO emission in the open ocean after the air mass passes through this site, thus highlighting
the strong influences of the urban outflow to the downward marine areas in terms of
nighttime chemistry. During the CAM period, nocturnal ozone was higher, while $NO_x$ was
much lower. The $NO_3$ production was still very fast, with a rate of 1.2 ppbv h$^{-1}$. With the
absence of $N_2O_5$ measurement in this period, the $NO_3$ reactivity towards VOCs and $N_2O_5$
uptake were calculated to assess $NO_3$ loss processes. We showed that the average k($NO_3$)
from VOCs (56.5%, $2.6 \pm 0.9 \times 10^{-3}$ s$^{-1}$) was higher than $N_2O_5$ uptake (43.5%, $2.0 \pm 1.5 \times$
$10^{-3}$ s$^{-1}$) during the CAM period, indicating a longer $NO_3/N_2O_5$ lifetime than that during
IAM period. This study improves the understanding of the nocturnal $NO_3$ budget and
environmental impacts with the interaction of anthropogenic and natural activities in
marine regions.

**1. Introduction**
Reactive nitrogen compounds, especially the nitrate radical ($NO_3$) and dinitrogen pentoxide
($N_2O_5$) play an essential role in nocturnal atmospheric chemistry (Wayne et al., 1991;
Brown and Stutz, 2012). $NO_3$ is mainly generated via the oxidation of $NO_2$ by $O_3$ (R1), and
then $NO_3$ further reacts with $NO_2$ to produce $N_2O_5$ (R2) with a thermal equilibrium. The
temperature-dependent equilibrium constant, $K_{eq}$, regulates the equilibrium favoring $NO_3$
and $NO_2$ at higher temperatures (Osthoff et al., 2007; Chen et al., 2022). During daytime,
the $NO_3$ mixing ratio is generally low as its lifetime is very short (< 5 s) due to the fast
photolysis (R3) and rapid reaction with NO (R4) (a rate constant of $2.6 \times 10^{-11}$ cm$^3$
molecule$^{-1}$ s$^{-1}$ at 298 K; Atkinson et al., 2004). While at night, $NO_3$ accumulates and can
reach tens to hundreds of parts per trillion volume (pptv), making it the major nocturnal
oxidizing agent (Wang et al., 2015).
$NO_2 + O_3 \rightarrow NO_3 + O_2$     (R1)
$NO_2 + NO_3 \rightleftharpoons N_2O_5$     (R2)
$NO_3 + h\nu \rightarrow NO_2 + NO$     (R3)
$NO + NO_3 \rightarrow 2NO_2$     (R4)





During nighttime, $NO_3$ is the most important oxidant for alkenes (Mogensen et al., 2015;
Edwards et al., 2017), particularly in rural, remote, or forested environments, where it
predominantly reacts with unsaturated biogenic volatile organic compounds (VOCs),
especially isoprene and monoterpene (Ng et al., 2017; Liebmann et al., 2018b; Liebmann
et al., 2018a), to form alkyl nitrates ($RONO_2$), that ultimately lead to secondary organic
aerosols (SOAs) (Brown and Stutz, 2012). The observations and model simulations showed
that the measured particulate organic nitrates were largely attributed to the nocturnal $NO_3$
oxidation across Europe (Kiendler-Scharr et al., 2016). The $NO_3$ oxidation was also
reported to play an important role in aerosol formation in the Southeastern United States
with high isoprene and monoterpene emissions (Xu et al., 2015). These studies highlighted
the critical role of the reaction of $NO_3$ with VOCs in $NO_3$ budget and organic aerosol
pollution. In addition, $NO_3$ also reacts with dimethyl sulfide (DMS) over the ocean,
affecting the marine sulfur cycle and thus cloud formation and global climate (Aldener et
al., 2006; Brown and Stutz, 2012; Ian Barnes et al., 2006; Rosati et al., 2022). While in
high aerosol loading regimes, the $N_2O_5$ heterogeneous uptake becomes a significant
indirect $NO_3$ loss pathway. The hydrolysis reaction produces nitrate ($NO_3^-$) and nitryl
chloride ($ClNO_2$) on chloride-containing aerosols surfaces (Osthoff et al., 2008; Thornton
et al., 2010), in which $ClNO_2$ activates the Cl radical and further enhances the
photochemistry and ozone pollution in the following day (Riedel et al., 2012; Riedel et al.,
2014; Behnke et al., 1993).
Different $NO_3$ loss pathways produce different air pollutants and cause different
environmental impacts, characterization of $NO_3$ budget is essential to clarify the $NO_3$
chemistry in air pollution under various environments. Observations of $N_2O_5$ and $NO_3$ in
different regions and evaluation of their loss processes have been reported in numerous
studies (Crowley et al., 2011; Geyer et al., 2001; Brown et al., 2011; Dewald et al., 2022;
Niu et al., 2022; Brown et al., 2016; Wang et al., 2020a; Tham et al., 2016; Aldener et al.,
2006; Lin et al., 2022). In general, the $NO_3$ loss process shows significant regional
differences. In urban areas featuring intensive anthropogenic $NO_x$ emissions and moderate
(or high) aerosol loading, $N_2O_5$ uptake is comparable or even dominates the $NO_3$ loss
(Wang et al., 2013). While in rural and forested areas with abundant biogenic VOCs
(BVOC) emissions, the $NO_3$ loss processes were usually dominated by BVOCs (Dewald
et al., 2022; Geyer et al., 2001; Brown et al., 2011). As for the coastal areas, which were
jointly affected by the polluted air mass from the inland and the relatively clean air mass
from the ocean, the dominant $NO_3$ loss process varies greatly depending on the air mass
origin (Aldener et al., 2006; Niu et al., 2022; Brown et al., 2016; Crowley et al., 2011). For
instance, Crowley et al. (2011) found in the Atlantic coast of Southern Spain (forested area)
that when the air mass mainly originated from the Atlantic, $NO_3$ was mainly consumed by
BVOCs (mainly monoterpenes) emitted from nearby forests, while when the air mass came



from the continent, NO$_3$ loss was mainly due to reactions with anthropogenic VOCs
(AVOCs).
China has been recently proven to be a hot spot of nocturnal chemistry with a high NO$_3$
production rate (Wang et al., 2023). Many studies have reported the mechanisms, budget,
or impacts of NO$_3$-N$_2$O$_5$ chemistry in different regions, while most of them were conducted
in urban regions (Wang et al., 2013; Yan et al., 2021; Wang et al., 2020a; Wang et al., 2017c;
Wang et al., 2017d). For example, Wang et al. (2017b) showed a significant contribution
of N$_2$O$_5$ uptake to nitrate pollution in summer and winter, and they also highlighted the fast
organic nitrate production rate observed in Beijing rural region in summer (Wang et al.,
2018b). Only several studies focused on nighttime oxidation in coastal cities like Shanghai,
Shenzhen, and Hong Kong (Zhu et al., 2022; Niu et al., 2022; Yan et al., 2019), which
showed different patterns of NO$_3$ chemistry compared with urban regions. Even fewer field
studies were conducted on the island which is far away from the coastal cities where the
interactions of the oceanic atmosphere and urban plumes can significantly affect the NO$_3$
budget and impacts. Given the diversity of air masses in inland and coastal areas, studies
are needed to gain a comprehensive understanding of NO$_3$ losses in different atmospheric
environments, particularly in coastal and marine areas.
Therefore, we conducted an intensive field observation on Da Wan Shan Island (DWS) in
the winter of 2021, which is a typical island site in the north of the South China Sea, and
downward of the city clusters in the Pearl River Delta, China during the winter monsoon
periods. The island features a subtropical oceanic monsoon climate, and the north and
northeast synoptic winds from inland PRD and eastern China coast are generally
predominant in winter (Liu et al., 2019; Wang et al., 2018a). This allows us to further
investigate the interactions between anthropogenic emissions and marine emissions from
the perspective of nighttime chemistry. In this study, the measurements of N$_2$O$_5$ and the
related species observed during the DWS winter campaign are reported. We have identified
two types of air masses from both mainland China and coastal areas. Finally, the NO$_3$
budget and loss processes in different air masses are characterized.
**2.   Methods**
**2.1 Site description**
The field campaign was conducted at Da Wan Shan Island (21.93° N, 113.72° E) from Nov.
9$^{th}$ to Dec. 16$^{th}$, 2021. Fig. 1 shows the study site, which is approximately 60 km southwest
of Hong Kong; 40 km southeast of Zhuhai; and about 100 km and 80 km away from the
megacities Guangzhou and Shenzhen, respectively. This island is dominated by





mountainous terrain with an area of 8.1 km$^2$ and has a small population of about 3000.
Anthropogenic emissions are sparse and no industrial pollution sources were identified,
though numerous ships engaging in local fishing activities were observed, potentially
affecting the local atmosphere. During the measurement, local airflow was consistent from
the northwest to southeast (Fig. 1a) due to the winter monsoon, with wind speeds most
frequently ranging from 1.8 to 7.9 m s$^{-1}$ (10$^{th}$ - 90$^{th}$ percentiles) and an average of 4.5 ± 2.6
m s$^{-1}$. This wind direction is indicative of the mixing of air masses from both continental
and coastal areas. The HYbrid Single-Particle Lagrangian Integrated Trajectory model
(HYSPLIT) was adopted to investigate the historical trajectory. The HYSPLIT model was
run for 48 hours backward in time at local times of 20:00, 24:00, and 04:00, and at a height
of 70 m above sea level. It confirmed that the airmass during nighttime mostly came from
continental China (defined as inland air mass, IAM, 84%) and the coastal areas (defined as
coastal air mass, CAM, 16%). No air masses free of pollution from the South China Sea
were observed during the measurement period. All measurement instruments were placed
in the DWS Atmospheric-Marine Research Station, located on the rooftop with inlets
approximately 10 m above ground level and about 72 m above sea level. All times were
given in CNST (Chinese National Standard Time = UTC + 8 h), with sunrise around 06:40
CNST and sunset at 17:40 CNST.

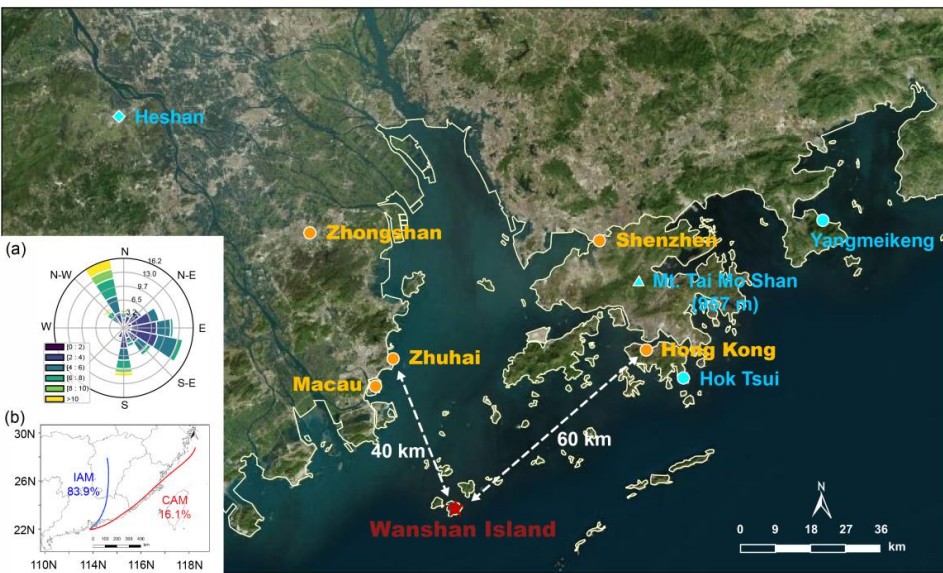


**Figure 1.** A map of the field measurement site of Wanshan Island (red star) and the
surrounding environment (extracted from BingSatelliteMap). Two coastal sites Hok Tsui
(Yan et al., 2019) and Yangmeikeng (Niu et al., 2022), and an urban site Heshan (Wang et




al., 2022; Yun et al., 2018b) are denoted as blue circle and diamond, respectively. The blue
triangle denoted Mt. Tai Mo Shan (957 m), a mountainous site that studied the nighttime
chemistry in the nocturnal residual layer (Brown et al., 2016). The inset plot (a) provides
the wind rose for the sampling site during the campaign. Panel (b) shows the clustering
result of the 48-hrs backward trajectory calculations using the HYSPLIT model throughout
the campaign.
**2.2 Instrument setup**
Various parameters were measured in this study, including $N_2O_5$, NO, $NO_2$, $O_3$, VOCs,
particle number size distribution (PNSD), and meteorological parameters with different
instruments. The detail information about these instruments is listed in Table 1. The $N_2O_5$
measurements were performed using a cavity-enhanced absorption spectrometer (CEAS)
which has been deployed in several field campaigns (Wang et al., 2017a; Wang et al., 2017b;
Wang et al., 2018b; Wang et al., 2020b). In brief, ambient $N_2O_5$ was thermally decomposed
to $NO_3$ in a perfluoro alkoxy alkane (PFA) tube (length: 35 cm, I.D.: 4.35 mm) heated to
130 °C, and $NO_3$ was detected within a 110 °C PFA resonator cavity. NO was injected to
destroy $NO_3$ from $N_2O_5$ thermal decomposition every 5 min cycle, and the result was used
as the reference spectrum to avoid the influence of ambient water vapor. A Teflon
polytetrafluoroethylene (PTFE) filter was used to remove ambient aerosol particles, and
the inlet flow rate was 1.0 L·min$^{-1}$. The loss of $N_2O_5$ in the sampling line and filter was
considered in the data correction. The limit of detection (LOD) was 2.7 pptv (1σ), and the
measurement uncertainty was ± 19%.
**Table 1.** The information of observation instruments used during the DWS campaign.

| Species | Techniques | Detection limit | Accuracy | Time resolution |
|---------|-----------|-----------------|----------|-----------------|
| $N_2O_5$ | CEAS | 2.7 pptv (1σ) | ±19% | 10 s |
| NO | Chemiluminescence | 0.4 ppbv | ±5% | 1 min |
| $NO_2$ | Chemiluminescence | 0.4 ppbv | ±5% | 1 min |
| $O_3$ | UV photometry | 0.4 ppbv | ±5% | 1 min |
| VOCs | PTR-TOF-MS | 0.01 ppbv | ±10% | 10 s |
| PNSD | SMPS | 5–300 nm | ±10% | 5 min |

$NO_x$ and $O_3$ were measured by commercial instruments (model T200U and model T400U,
Teledyne API Inc., respectively) calibrated with zero air before the measurement. The
nitrogen oxide analyzer uses the chemiluminescence detection method to measure the
original NO and converted $NO_2$. Aerosol surface area density ($S_a$, $\mu m^2\ cm^{-3}$) was calculated
based on the particle numbers and geometric diameter, which was measured by a





laboratory-assembled scanning-mobility particle sizer (SMPS). This SMPS system consists
of two differential mobility analyzers (DMA, "nano-DMA" mode 3081A, and "regular-
DMA" mode 3085A, TSI Inc.) in parallel, and a condensed particle counter (mode 3787,
TSI Inc.) as the detector. The combination of nano DMA and conventional mode 3085A
DMA enables the SMPS to have better detection performance for particles below 50 nm.
In this measurement, SMPS measured the particle size distribution in 5-300 nm with a time
resolution of 5 minutes, and Sa can be regarded as the lower limit value. A growth factor
$f(RH) = 1 + 8.8 \times (RH/100)^{9.7}$ (Liu et al., 2013) was used here to correct dry state $S_a$ to wet
state $S_a$.
VOCs were measured by proton transfer reaction time of flight mass spectrometry (PTR-
TOF-MS, Ionicon Analytik GmbH, Innsbruck, Austria) with a time resolution of 10 s.
Meanwhile, VOCs were also sampled every 2 h using 2 L canisters on the selected days
when the hourly $O_3$ mixing ratio exceeded 70 ppbv, and the canister samples were analyzed
by a gas chromatograph equipped with a mass spectrometer or flame ionization detector
(GC-MS). For the absence of nocturnal data from canister samples, the following analysis
was based on the PTR-TOF-MS measurement. Since monoterpene species cannot be
distinguished by PRT-TOF-MS, the reaction rate constant of the sum monoterpene with
$NO_3$ was weighted by the campaign-averaged percentage of α-pinene and β-pinene
detected by GC-MS. Meteorological parameters (i.e., temperature (T), relative humidity
(RH), wind speed, and wind direction) were routinely monitored with a time resolution of
5 min.
**2.3 The calculation of $NO_3$ budget and lifetime**
With the observation of $N_2O_5$, $NO_3$ can be calculated according to their temperature-
dependent equilibrium relationship (Eq. 1) (Brown and Stutz, 2012). The production rate
of nitrate radical, P($NO_3$), is commonly expressed by Eq. 2, where $k_{NO_2+O_3}$ represents the
temperature-dependent reaction rate constant of $NO_2$ and $O_3$ (Atkinson et al., 2004). In
general, the nocturnal $NO_3$ losses typically include three main pathways: (1) the reaction
with NO, (2) the reactions with VOCs, and (3) $N_2O_5$ uptake.
$[NO_3] = [N_2O_5]/Keq(T)[NO_2]$
$Keq = 5.50 \times 10^{-27} \times \exp(10724/T)$     (Eq. 1)
$P(NO_3) = k_{NO_2+O_3}[O_3][NO_2]$    (Eq. 2)
$L(NO_3) = \sum k_i[VOC_i][NO_3] + k_{NO+NO_3}[NO][NO_3] + k_{het}[N_2O_5]$   (Eq. 3)



The $NO_3$ reactivity towards VOCs, $k(NO_3)$, is the first-order loss rate coefficient
calculated from the products of the bimolecular rate coefficients $k_i$ and the VOC
concentrations as shown in Eq. 4.
$k(NO_3) = \sum k_i [VOC_i]$     (Eq. 4)
The $k_{het}$ is the first-order loss rate coefficient of $N_2O_5$ uptake on the aerosol surface. It
depends on the uptake coefficient $\gamma(N_2O_5)$, the aerosol surface area density Sa ($\mu m^2$ $cm^{-3}$),
and the mean molecular speed $c$ (Eq. 5). The $\gamma(N_2O_5)$ is influenced by chemical
composition, physical properties of aerosol, as well as ambient conditions including related
humidity and temperature (Yu et al., 2020; Wagner et al., 2013; Wang et al., 2018b; Bertram
and Thornton, 2009; Tang et al., 2014; Kane et al., 2001). Here $\gamma(N_2O_5)$ is parameterized
based on RH and temperature (Eq. 6) (Hallquist et al., 2003; Kane et al., 2001; Evans and
Jacob, 2005).
$k_{het} = \frac{1}{4} c S_a \gamma(N_2O_5)$     (Eq. 5)
$\gamma(N_2O_5) = \alpha \times 10^\beta$
$\alpha = 2.79 \times 10^{-4} + 1.3 \times 10^{-4} \times RH - 3.43 \times 10^{-6} \times RH^2 + 7.52 \times 10^{-8} \times RH^3$
$\beta = 4 \times 10^{-2} \times (T - 294)$   $(T > 282K)$
$\beta = -0.48$   $(T < 282K)$     (Eq. 6)
Lifetimes are commonly expressed as the ratio of their concentrations to the $NO_3$
production rate as determined by Eq. (7) and Eq. (8), assuming the production and loss are
in dynamic balance at night (Brown et al., 2003; Brown and Stutz, 2012).
$\tau_{N_2O_5} = \frac{[N_2O_5]}{P(NO_3)} = \frac{[N_2O_5]}{k_{NO_2+O_3}[NO_2][O_3]}$     (Eq. 7)
$\tau_{NO_3} = \frac{[NO_3]}{P(NO_3)} = \frac{[NO_3]}{k_{NO_2+O_3}[NO_2][O_3]}$     (Eq. 8)
**3. Results and discussion**
**3.1 Measurement overview**



The time series of N₂O₅, related trace gases, and selected meteorological parameters for
the study period are depicted in Fig. 2. The air masses are categorized into IAM and CAM
according to the backward trajectories at 20:00, 00:00, and 04:00 each day as illustrated in
Fig. 1. The detailed information of two kinds of air masses is listed in Table 2. Data gaps
for N₂O₅ were caused by technical problems, mirror reflectivity calibration, or instrumental
maintenance, which usually took place in the daytime. In this campaign, meteorological
conditions featured a typical subtropical winter climate with average temperature and RH
values of $20.1 \pm 1.9$ °C and $52.0\% \pm 13.6\%$, respectively.

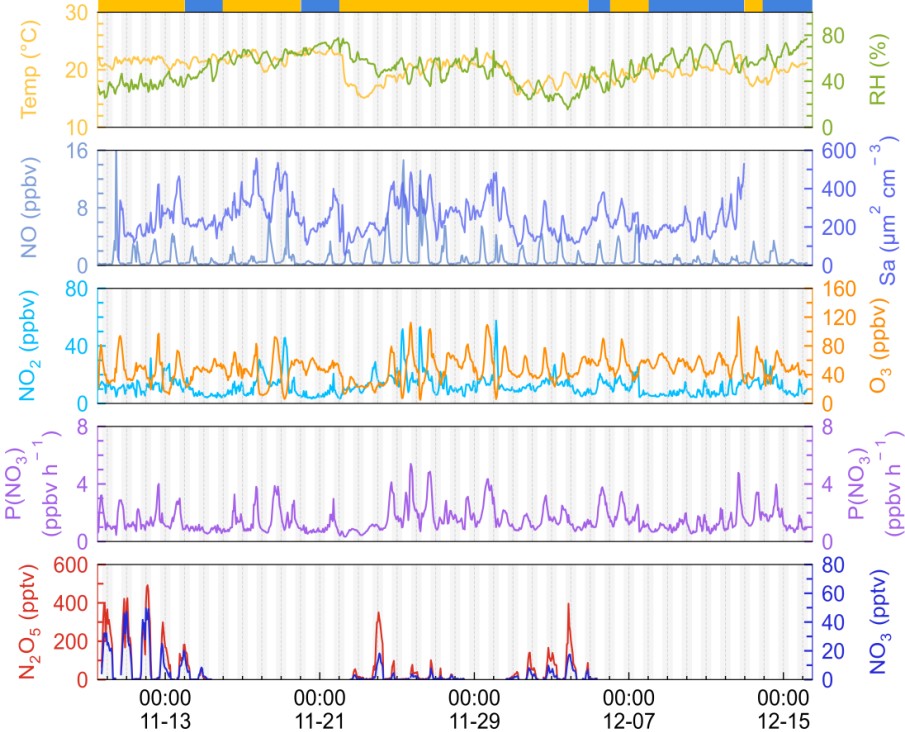


**Figure 2.** Time series of $N_2O_5$, $NO_3$, $P(NO_3)$, NO, $NO_2$, Sa, temperature, and relative
humidity in 1-hour average. The light gray shadow indicates the nighttime period. The
ribbon at the top separates the air masses into two categories, yellow for inland air masses
(IAM) and blue for coastal air masses (CAM).

Ozone exhibited the characteristic of afternoon photochemical peaks especially when the
airmass comes from the inland. The average and maximum concentrations of ozone were
$48.2 \pm 18.2$ ppbv and 120.1 ppbv, respectively, among which the hourly maximum level
exceeded the Chinese national air quality standard (200 μg m$^{-3}$, equivalent to 93 ppbv) for
6 days out of 37 days of measurements. All these $O_3$ pollution episode days occurred during
the IAM period. Meanwhile, the mixing ratio of NO, $NO_2$, and Sa usually increased during



these days, indicating that this site was strongly affected by regional transport from inland
China, i.e., the PRD region. Previous observations by Wang et al. (2018a) also found high
$O_3$ levels in autumn on the same island due to the weak NO titration and high $O_3$ production
rate. Detailed analysis of ozone and VOCs will be subjected to another manuscript (Fang
et al., 2023).
Nitrogen oxides ($NO_x = NO + NO_2$) were at a moderate level with an average value of 13.1
$\pm$ 8.2 ppbv, which is much lower than the values in PRD regions (usually > 20 ppbv, Wang
et al., 2022; Yang et al., 2022; Yun et al., 2018b) and higher than those on the remote islands
in South China Sea (< 5 ppbv, Chuang et al., 2013). The mixing ratio of NO at nighttime
was low and showed small peaks during daytime. With the $O_3$ accumulating throughout
the day, NO decreased to below the instrument detection limit in the first half of the night,
while it began to increase as the O3 concentration decreased in the second half of the night.
Given that the lifetime of NO is only a few minutes in the presence of several tens of ppbv
of $O_3$ (Dewald et al., 2022), NO is likely to come from a local source such as soil emission.
**Table 2.** Summary of detailed information on the two air mass types (mean $\pm$ standard
deviation).

| Species | IAM | | CAM | |
|---|---|---|---|---|
| | All day | Nighttime | All day | Nighttime |
| $O_3$ (ppbv) | 45.8 $\pm$ 20.2 | 42.9 $\pm$ 18.4 | 53.1 $\pm$ 11.9 | 51.4 $\pm$ 9.6 |
| $NO_x$ (ppbv) | 15.1 $\pm$ 8.7 | 14.5 $\pm$ 9 | 9.2 $\pm$ 5.1 | 8.8 $\pm$ 4.8 |
| $NO_2$ (ppbv) | 13.9 $\pm$ 7.6 | 14.1 $\pm$ 8.3 | 8.6 $\pm$ 4.8 | 8.6 $\pm$ 4.8 |
| NO (ppbv) | 1.2 $\pm$ 2.3 | 0.4 $\pm$ 1.1 | 0.5 $\pm$ 0.6 | 0.2 $\pm$ 0.1 |
| Temp (℃) | 19.9 $\pm$ 2 | 19.9 $\pm$ 1.9 | 20.8 $\pm$ 1.5 | 20.6 $\pm$ 1.5 |
| RH (%) | 46.7 $\pm$ 12.5 | 47.7 $\pm$ 13.2 | 61.2 $\pm$ 10 | 64.1 $\pm$ 9.6 |
| $P(NO_3)$ (ppbv $h^{-1}$) | 1.6 $\pm$ 0.9 | 1.5 $\pm$ 0.8 | 1.3 $\pm$ 0.8 | 1.2 $\pm$ 0.6 |
| $N_2O_5$ (pptv) | - | 119.5 $\pm$ 128.6 | - | - |
| $NO_3$ (pptv)[a] | - | 9.9 $\pm$ 12.5 | - | - |
| $\tau_{N_2O_5}$ (min) | - | 6.5 $\pm$ 6.5 | - | - |
| $\tau_{NO_3}$ (min) | - | 0.5 $\pm$ 0.7 | - | - |

Note: [a] $NO_3$ is calculated by the thermal equilibrium between $NO_2$, $NO_3$, and $N_2O_5$.
$N_2O_5$ was at a moderate level on most days with a nocturnal average of 119.5 $\pm$ 128.6 pptv,
with high concentrations (>400 pptv in 1-hour average) only lasting for three days. During
the nights from November 9th to 12th, the $N_2O_5$ concentrations were significantly higher
than those at other nights, with a maximum of 657.3 pptv at midnight of November 12th.
The $NO_3$ concentration (calculated based on the thermal equilibrium with $N_2O_5$) was also
moderate with an average mixing ratio of 9.9 $\pm$ 12.5 pptv, which was higher than that





reported on a nearby coastal site of Hong Kong Island (Yan et al., 2019). Table 3 compares
the $N_2O_5$, $NO_3$, and $P(NO_3)$ found in other coastal (or island) and continental regions from
Europe, the United States, and China. In our study, $N_2O_5$ and $NO_3$ were at a moderate level
compared to other coastal regions when they were affected by emission plumes from
continental regions, such as Northwestern Europe (Morgan et al., 2015), the East coast of
the USA (Brown et al., 2004), and Shenzhen, China (Niu et al., 2022), and were comparable
with urban regions (Wang et al., 2017b; Wang et al., 2018b). The concentrations of $NO_3$
precursors ($NO_2$ and $O_3$) at this site were much similar to some rural areas, leading to a
high $NO_3$ production rate with a daily average of $1.5 \pm 0.9$ ppbv h$^{-1}$ and a maximum of 5.9
ppbv h$^{-1}$. The average value is much higher than that reported in Beijing in winter (0.4 ppbv
h$^{-1}$,(Wang et al., 2021), comparable to autumn ($1.4 \pm 1.7$ ppbv h$^{-1}$, (Wang et al., 2017b) and
even higher than that in summer Taizhou ($1.01 \pm 0.47$ ppbv h$^{-1}$, (Wang et al., 2020a). The
nocturnal $P(NO_3)$ was $1.4 \pm 0.7$ ppbv h$^{-1}$, even higher than the average value in the warm
season of China with $1.07 \pm 0.38$ ppbv h$^{-1}$ (Wang et al., 2023). Besides the high $NO_2$ and
$O_3$, the high reaction rate constant for $NO_2$ and $O_3$ due to the high temperature at this site
is a potential explanation for the high $P(NO_3)$ values observed in this study. The high $P(NO_3)$
and the low concentrations of $N_2O_5$ and $NO_3$ indicate intensive atmospheric oxidation
capacity and fast $NO_3$ and $N_2O_5$ removal over the Pearl River Estuary.
The difference of trace gases in IAM and CAM periods, and the mean diurnal profiles of
$N_2O_5$, together with relevant species are shown in Fig. 3. Daytime $N_2O_5$ and $NO_3$ in the
IAM period were shown as zero due to the absence of observation. Because of limited $N_2O_5$
data for the CAM period, neither $N_2O_5$ nor $NO_3$ is shown in Fig. 3. NO exhibited similar
diurnal variation in both periods and the mixing ratio was higher in the IAM period. The
wind rose plot (Fig. S1) showed high concentrations of NO originating from the north
characterized by the outflow from PRD regions. However, $NO_2$ differed in the two periods,
showing highly anti-correlation with $O_3$ only in the IAM period and little diurnal variation
in the CAM period. A fit of nocturnal $O_3$ against $NO_2$ (Fig. S2) yields a slope of -1.1 in
IAM, implying that the major emission of $NO_x$ was NO and almost no nocturnal $NO_2$
production occurred (Brown et al., 2016).

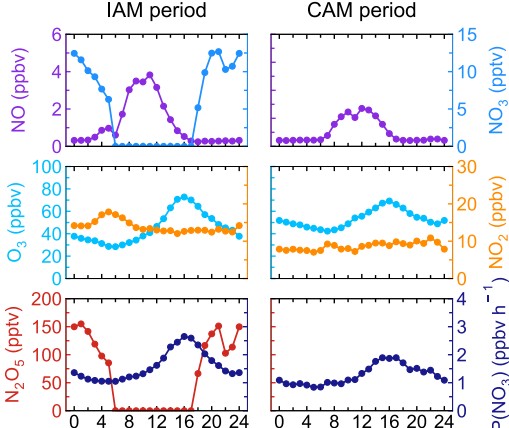

**Figure 3.** Mean diurnal profiles of $N_2O_5$, $P(NO_3)$, and relevant parameters in the two types of air masses.

Ozone exhibited a typical diurnal pattern for all airmasses, gradually increasing until its peak at 16:00 and then slowly decreasing throughout the night until its lowest mixing ratio was reached at about 06:00. Compared to the CAM period, the lower minimum hourly $O_3$ concentration and a small peak of $NO_2$ in the early morning indicated that NO titration effect was stronger in the IAM period, and the higher maximum of $O_3$ concentration in IAM indicated that photochemical formation of $O_3$ and/or transport was faster to completely offset the titration. In addition, the higher $NO_x$ and VOC concentrations in the IAM period facilitated $O_3$ formation. With the elevated precursor concentrations ($NO_2$ and $O_3$) in the IAM period, $N_2O_5$ and $NO_3$ accumulated rapidly after sunset, reaching their peak values (155.0 pptv for each) near 20:00. $P(NO_3)$ was highly consistent with $O_3$ in diurnal variation and reached the peak at 16:00, with peak values of 2.7 ppbv h$^{-1}$ (IAM) and 1.9 ppbv h$^{-1}$ (CAM), as well as a nocturnal average value of $1.5 \pm 0.8$ ppbv h$^{-1}$ (IAM) and $1.2 \pm 0.6$ ppbv h$^{-1}$ (CAM), respectively. The $P(NO_3)$ of CAM was consistent with the observation in eastern Shenzhen ($1.2 \pm 0.3$ ppbv h$^{-1}$) (Niu et al., 2022) during which the air masses were transported from clean areas or the sea surface.





333

**Table 3.** Summary of field-observed $N_2O_5$, $NO_2$, $O_3$ concentrations, and $NO_3$ production rate.

| Region | Location | Time | $N_2O_5$ (pptv) | $NO_3$ (pptv) | $NO_2$ (ppbv) | $O_3$ (ppbv) | $P$ ($NO_3$) (ppbv hr⁻¹) | References |
|---|---|---|---|---|---|---|---|---|
| Urban | Jinan, China | Aug.-Sep.,2014 | 22 ± 13 (max 278) | / | 74.6 | 55 | / | (Wang et al., 2017c) |
| Urban | Shanghai, China | Aug.-Oct., 2011 | 310 ± 380 | 16±9 (max 95) | 0-76 | 23 ± 8 (max 57) | 1.10 ± 1.09 | (Wang et al., 2013) |
| Urban | Beijing, China | May-Jul., 2016 | 100–500 (max 937) | 27 | / | / | 1.2 ±0.9 | (Wang et al., 2018b) |
| Urban | Mt. Tai, China | Jul.-Aug., 2014 | 6.8±7.7 | / | 16.4 (±6.1) | 88.6 (±16.6) | 0.45±0.40 | (Wang et al., 2017d) |
| Urban | Heshan, China | Sep.-Nov., 2019 | 64 ± 145 (night) (max 1180) | max 90 | 21.0±10.4 | 75.2±20.9 (max 152.8) | 2.5 ± 2.1 (day) 1.8 ±1.5 (night) | (Wang et al., 2022) |
| Urban | Beijing, China | Sep.-Oct., 2019 | 68.0±136.7 | / | 35.1 ±16.6 | 27.7 ±25.2 | 2.25 ±2.02 | (Wang et al., 2017b) |
| Suburban | Changzhou, China | May-Jul., 2019 | 53.4 ± 66.1 (max 304.7) | 4.7 ± 3.5 (max 17.7) | 13.7 ± 8.9 | 48.4 ± 27.8 | 1.7 ± 1.2 (max 7.7) | (Lin et al., 2022) |
| Rural | Wangdu, China | Jun.-Jul., 2014 | <200 (max 430) | / | 10-80 | (max 146) | 1.7 ±0.6 | (Tham et al., 2016) |
| Rural | Taizhou, China | May-Jul., 2018 | 26.0 ± 35.7 (max 492) | 4.4 ± 2.2 (max 150) | 28.28 ± 18.57 | 48.2 ± 32.5 | 1.01 ± 0.47 (night) | (Wang et al., 2020a) |
| Coastal | Tai Mo Shan, HK | Nov.-Dec., 2013 | 0.5-11.8 ppbv | - | 7.88 | 68.5 | 0.01-2 | (Brown et al., 2016) |
| Coastal | East coast of USA | Jun.-Aug., 2002 | 85 | 17 | 6 | 35 | / | (Brown et al., 2004) |
| Coastal | California, USA | Jan.,2004 | 0-200 | / | 0-15 | 15-35 | / | (Wood et al., 2005) |
| Coastal | Southern Spanish | Nov.-Dec., 2018 | ~500 (max) | / | 1-15 | 15-40 | / | (Crowley et al., 2011) |
| Coastal | Shenzhen, China | Sep.-Oct., 2019 | 55.6 ± 89 (max 1420) 45.4 ± 55.2 (BAM) | / | 6.2 | 88.9±24.6 | 2.9 ± 0.5 (UAM) 1.2 ± 0.3 (BAM) | (Niu et al., 2022) |
| Coastal | Northwestern Europe | Jul., 2010 | 670 | / | 0.5-2 | 30-40 | / | (Morgan et al., 2015) |
| Island | Hok Tsui, HK | Aug.-Sep., 2012 | 17±33 (max 336) | 7 ± 12 | 6 ± 7 | 33 ± 24 | / | (Yan et al., 2019) |
| Island | Wanshan, China | Nov.-Dec., 2021 | 107.22 ± 125.17 | 7.56 ± 10.95 | 13.14 ± 8.68 | 43.75 ± 18.49 | 1.38 ± 0.83 | This work |

Notes: UAM means air masses coming from continental areas, and BAM means air masses coming from background marine areas.
Mean values are in the form of mean ± standard deviation or single data. The maximum was noted in the table.





## 3.2 The lifetimes of N₂O₅ and NO₃

Steady-state lifetime is one of the most common and useful diagnostics for $NO_3$ and $N_2O_5$ analysis in the atmosphere (Brown et al., 2003; Wang et al., 2018b; Wang et al., 2020a; Brown et al., 2016). As shown in Fig. 4, $\tau_{NO_3}$ was low during the whole campaign with an average of 0.5 ± 0.7 min. $\tau_{N_2O_5}$ showed a similar pattern to $\tau_{NO_3}$ but had a much higher value, ranging from 0 to 34.1 min with an average of 6.1 ± 6.5 min. The $N_2O_5$ lifetime was higher in the first half of the campaign (11.5 min, November 9[th] to 14[th]) than in the second half (3.5 min, November 22[th] to 28[th]). The difference was mainly due to the $N_2O_5$ mixing ratio rather than P($NO_3$), as P($NO_3$) shows no significant difference during the whole observation (Fig. 2).

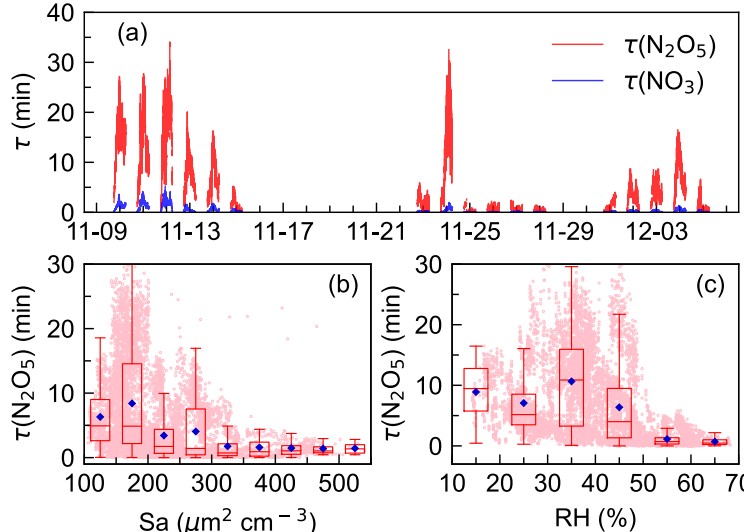

**Figure 4.** Time series of $N_2O_5$ and $NO_3$ lifetimes (a) and variations of nocturnal $N_2O_5$ lifetime as a function of aerosol surface area density, Sa (b), and relative humidity, RH (c). The blue diamond represents the average $\tau_{N_2O_5}$ and pink dots represent the scatter data point in 1 min.

$\tau_{N_2O_5}$ values were comparable to those measured on the coastline of Finokalia, Greece for a median of 5 min (Vrekoussis et al., 2004; Vrekoussis et al., 2007), but much lower than those previously reported in the residual layer in Hong Kong for 1-5 h (Brown et al., 2016). In comparison, the lifetimes were much longer than in inland urban areas, for example, 0.93 ± 1.13 min in Taizhou (Li et al., 2020), 1.6 ± 1.5 min in Changzhou (Lin et al., 2022) for YRD regions, 1.1-10.7 min (Zhou et al., 2018) and 4.5 ± 4.0 min (Wang et al., 2018b)



in Beijing. Typically, high aerosol loading, more intensive VOC, and NO emissions in these
areas led to enhanced $N_2O_5$ uptake and reactions of $NO_3$ with VOC. But at this site, the
atmosphere was relatively clean since the maximum $S_a$ value was less than 600 $\mu m^2$ $cm^{-3}$,
making $N_2O_5$ uptake slow. Fig. 4b shows $N_2O_5$ lifetime decreased rapidly from 8.3 min to
1.7 min when $S_a$ increased up to 300 $\mu m^2$ $cm^{-3}$ and then remained at relatively low constant
levels though $S_a$ still increased. Such a trend of $\tau_{N_2O_5}$-$S_a$ dependence was consistent with
previous observations and varied in exact values (Zhou et al., 2018; Wang et al., 2018b; Li
et al., 2020). Fig. 4c. showed that $\tau_{N_2O_5}$ decreased as RH increased ($>40\%$) possibly due
to the hygroscopic aerosol growth and the dependence of the $N_2O_5$ uptake coefficient on
the RH (Brown and Stutz, 2012). Overall, the trend is consistent with previous works, while
the large discrepancy of the dependence implied that $N_2O_5$ uptake was not the dominant
$NO_3$ loss process.
**3.3 The $NO_3$ reactivity and $N_2O_5$ uptake coefficients**
The concurrent high $P(NO_3)$ and low $NO_3$ lifetime imply high $NO_3$ reactivity as well as a
large nocturnal $NO_3$ loss process at DWS. The $NO_3$ reactivity ($k(NO_3)$) towards VOCs was
calculated by Eq. 4, towards which were categorized into anthropogenic VOC and biogenic
VOC. Throughout the campaign, $k(NO_3)$ varied considerably (Fig. 5a), showing relatively
high and fluctuated values when the airmasses featured IAM. The $k(NO_3)$ ranged from 1.6
$\times 10^{-3}$ $s^{-1}$ to $2.4 \times 10^{-2}$ $s^{-1}$ with the daily average of $4.6 \pm 2.8 \times 10^{-3}$ $s^{-1}$. Low values of $k(NO_3)$
were observed from December $9^{th}$ to $12^{th}$ when the air masses originate from coastal or
offshore from the east and southeast, which features the outflow of coastal cities like Hong
Kong and Shenzhen.
Fig. 5b shows the mean diurnal profile of $k(NO_3)$, where a trend of high values in the
daytime and low values at nighttime are observed. Anthropogenic VOC, especially cresol,
dominated the daily trend of $k(NO_3)$, while biogenic VOC-$k(NO_3)$ showed no significant
diurnal variation. Except cresol, other highly reactive VOC showed little change
throughout the day. Regarding the biogenic VOC-$k(NO_3)$, the concentrations of
monoterpene, isoprene, and DMS changed smoothly although their emissions would
increase with elevated temperature and sunlight during daytime (Fuentes. et al., 2000). The
detailed contributions of VOC categories to $k(NO_3)$ were shown in Fig. 5c. The $k(NO_3)$
was $5.6 \pm 2.8 \times 10^{-3}$ $s^{-1}$ and $3.7 \pm 2.5 \times 10^{-3}$ $s^{-1}$ on average for daytime and nighttime,
respectively. The daytime distribution of $k(NO_3)$ differed from that at the mountaintop of
Tai Mo Shan in Hong Kong (Brown et al., 2016). During the nighttime, anthropogenic
VOC-$k(NO_3)$ tripled the biogenic VOC-$k(NO_3)$ and was dominated by cresol (26.4%). The
nighttime $k(NO_3)$ corresponded to a $NO_3$ lifetime of 4.5 min, which was about 10 times the
lifetime derived from steady-state analysis, indicating that the reaction of $NO_3$ with VOC





was not significant enough. The faster NO₃ loss rate also indicated the less aged air mass
that was influenced by surface-level emissions.

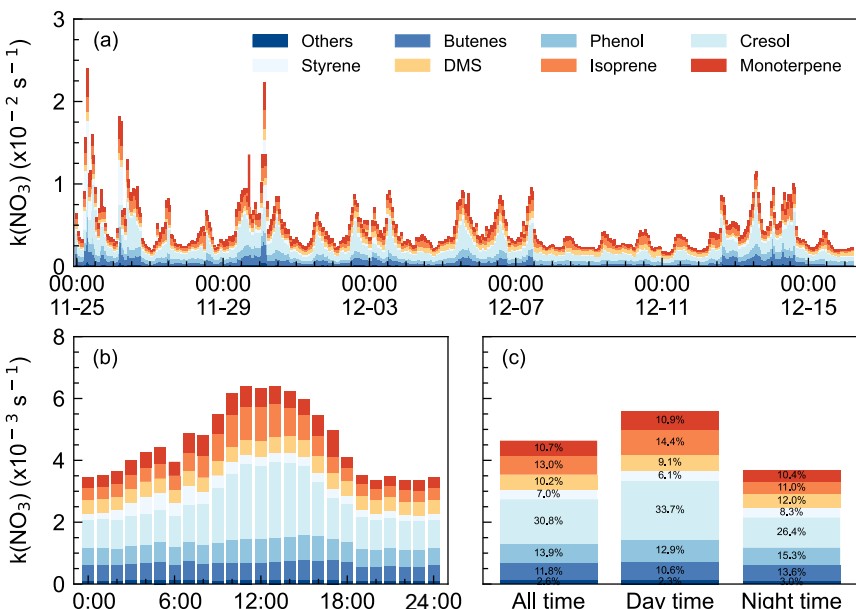


**Figure 5.** NO₃ reactivity via VOCs during the campaign. (a) k(NO₃) time series from Nov.
25ᵗʰ to Dec. 15ᵗʰ; (b) mean diurnal profiles; and (c) the relative contribution in different
categories.
We showed that NO₃ reactivity and its composition in this study exhibited significant
differences compared to other urban or forested regions (Wang et al., 2017d; Ayres et al.,
2015; Brown et al., 2016; Lin et al., 2022). Although anthropogenic VOCs played a
dominant role, accounting for 66.1%, the major contributors were not low-carbon alkenes
but phenol and cresol, which have received little attention in previous studies. Despite their
relatively low concentrations, averaging $7 \pm 3$ pptv and $4 \pm 3$ pptv respectively, their
substantial contribution to k(NO₃) is notable due to their fast rate constants ($3.8 \times 10^{-12}$ cm³
molecule$^{-1}$ s$^{-2}$ and $1.4 \times 10^{-11}$ cm³ molecule$^{-1}$ s$^{-2}$ at 298 K, respectively) for reaction with
NO₃. These substances are mainly secondary species from aromatic compounds and
significantly higher concentrations have also been observed in urban areas (Delhomme et
al., 2010; Zhu et al., 2005; Belloli et al., 1999). Hence, these phenolic compounds were
potentially important but often overlooked for their contributions to NO₃ reactivity in urban
areas, and their reactions with NO₃ may also contribute to the formation of nitrophenol.
These reactions warrant further attention in future research. Regarding biogenic VOCs,



besides the contributors commonly observed in forest regions such as monoterpenes and
isoprene, the marine emissions indicator, dimethyl sulfide (DMS), contributed 10.2% to
$NO_3$ reactivity (daily average). Previous studies have suggested that DMS may serve as a
major direct sink for $NO_3$ in clean marine regions (Allan et al., 1999; Aldener et al., 2006;
Brown et al., 2007). However, this study reveals that anthropogenic VOC emissions
significantly enhanced the $NO_3$ reactivity in marine areas, highlighting the crucial influence
of anthropogenic activities on marine atmospheric chemistry.
As shown in Fig. 6a, k($NO_3$) differed significantly between the inland and coastal air
masses, with $5.2 \times 10^{-3}$ $s^{-1}$ and $3.7 \times 10^{-3}$ $s^{-1}$ on average in IAM and CAM periods,
respectively. Of which anthropogenic VOC-k($NO_3$) in IAM ($3.5 \times 10^{-3}$ $s^{-1}$) was higher than
in CAM ($2.3 \times 10^{-3}$ $s^{-1}$) and dominant in both air masses, while biogenic VOC-k($NO_3$) was
comparable. The difference indicated that this region was affected by long-range transport
emissions to a certain extent. The pie charts in Fig. 6b showed different VOC categories
that contributed to k($NO_3$) in two periods with AVOC dominant at any time. The change in
the relative contribution of various VOCs to k($NO_3$) varied simultaneously throughout the
day, reflecting as butene, phenol, and DMS increased, while cresol and monoterpene
decreased from daytime to nighttime.
$N_2O_5$ heterogeneous uptake on aerosol is one of the vital loss processes of $NO_3$ and the
uptake coefficient varied greatly under different environmental conditions. For instance,
$\gamma(N_2O_5)$ can reach up to 0.072 in polluted urban regions (Wang et al., 2017b; Wang et al.,
2018b; Lu et al., 2022; Li et al., 2020), while usually below 0.03 in coastal areas (Brown
et al., 2016; Morgan et al., 2015; Niu et al., 2022). $N_2O_5$ uptake coefficient can be gotten
from the pseudo steady state method by assuming that $N_2O_5$ and $NO_3$ have achieved a
steady state (Brown et al., 2009), in which the fitted slope represents $\gamma(N_2O_5)$ and the
intercept represents the direct loss rate coefficient, k($NO_3$). However, this approach failed
to generate valid results in our study since a negative slope or intercept was observed (Fig.
S3). These results indicated that a large $NO_3$ removal process existed at this site, making it
unable to approach a stable state. Based on relative humidity and temperature, we
calculated the uptake coefficient by Eq.6 from November $9^{th}$ to $16^{th}$. The parameterized
average $\gamma(N_2O_5)$ showed a large variation ranging from 0.0014 to 0.0299, with an average
of 0.0095 ± 0.0059. This value is within the range from <0.0016 to 0.03 derived from the
ambient observation around other coastal areas (Niu et al., 2022; Yun et al., 2018a; Brown
et al., 2006; Brown et al., 2016; Morgan et al., 2015) and smaller than the polluted North
China Plain (Wang et al., 2017c; Wang et al., 2017b; Wang et al., 2017d; Tham et al., 2018).





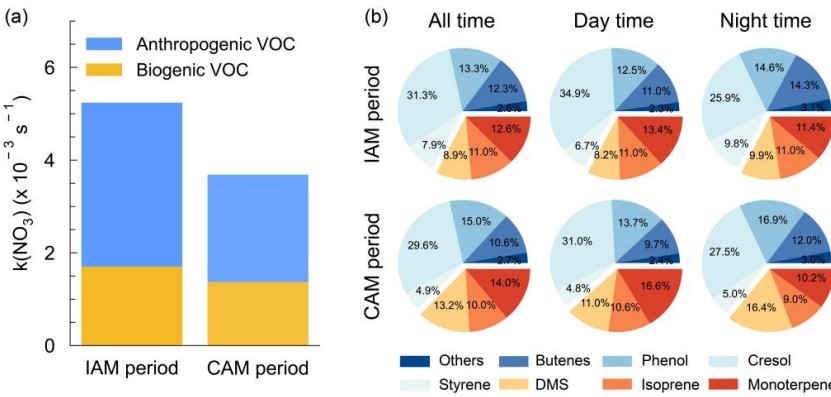


**Figure 6.** (a) Distributions of k(NO$_3$) in continental and coastal air masses. (b) The relative contribution of VOC categories to the k(NO$_3$).

### 3.4 The NO$_3$ loss budget

To assess the contribution of various loss processes to the total NO$_3$ removal, we calculated their loss rate and the loss ratio, $LR$(NO$_3$). Considering the short lifetime of NO$_3$, here the total NO$_3$ loss is represented by P(NO$_3$) which characterizes the atmospheric oxidation capacity of NO$_3$ to its reactants. Due to the data absence of measured VOCs or N$_2$O$_5$ during certain periods, the loss proportion of VOCs and N$_2$O$_5$ uptake in NO$_3$ loss only presented from Nov. 26th to Dec. 5th during which all air masses originated from continental China. As shown in Fig. 7, a closer examination revealed that the nights can be divided into two periods, period I: November 25th to 28th when the loss ratio of VOC and N$_2$O$_5$ uptake remained below 3%, and period II: November 30 to December 4 when the loss ratio was higher. Both periods had large nocturnal NO$_3$ production rates with an average of 2.1 ± 1.1 ppbv h$^{-1}$ in period I and 1.4 ± 0.6 ppbv h$^{-1}$ in period II, respectively.

N$_2$O$_5$ uptake rate was larger in period I (0.01 ± 0.01 ppbv h$^{-1}$) than that in period II (0.006 ± 0.004 ppbv h$^{-1}$), which can be explained by the increased RH, Sa, and N$_2$O$_5$ concentration as shown in Fig. 2. The loss ratio of these processes was shown in Fig. 7b, the total NO$_3$ loss through reactions with VOCs and N$_2$O$_5$ uptake accounted for less than 20%, with an average of 1.2% (period I) and 5.3% (period II), respectively. This result shows that the nighttime NO$_3$ chemistry may be almost negligible, compared with the NO$_x$ removal capacity during the day according to previous works reported in urban regions (Wang et al., 2017b; Wang et al., 2020a). The diurnal variation of the NO$_3$ loss fraction of both periods was shown in Fig. 7c and 7d, revealing that NO$_3$ loss via N$_2$O$_5$ uptake and VOCs




was slightly higher in the early evening and relatively stable in the late evening. The pie
charts in the center were the relative contribution between VOCs and $N_2O_5$ uptake, showing
that VOCs were overwhelming compared with $N_2O_5$ uptake during the two periods, with
an average of 68.4% and 91.7% during the first and second periods, respectively.

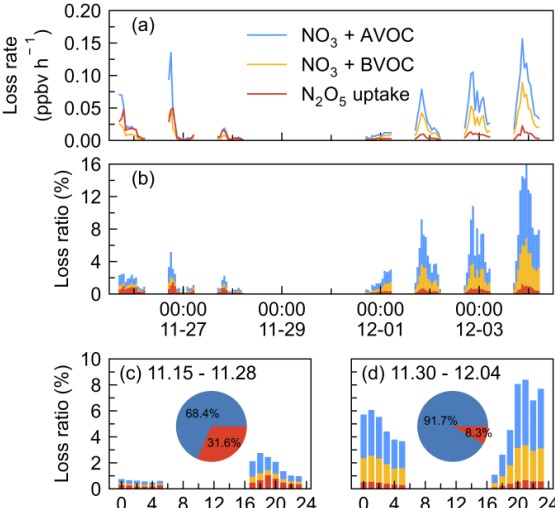

**Figure 7.** Time series of (a) the loss rate of $NO_3$ reactions with AVOC, BVOC, and $N_2O_5$
uptake, and (b) fractional contribution to the $NO_3$ loss during the nighttime by taking
$P(NO_3)$ as the total $NO_3$ loss in the IAM period. The mean diurnal profiles of $NO_3$ loss
ratio in two periods (c) November 25[th] - 28[th], and (d) November 30[th] - December 4[th]. Pie
charts in the center showed the relative contribution of VOCs (blue) and $N_2O_5$ uptake (red)
in $NO_3$ loss.
Due to the difficulty in experimental quantifying $\gamma(N_2O_5)$, the estimation of $N_2O_5$ uptake
in $NO_3$ loss may include some uncertainty. Considering the uncertainty both in
parameterized $\gamma(N_2O_5)$ and the $NO_3$ reactivity calculation, three sensitivity tests were
conducted to assess the uncertainty in period II because of the relatively high loss ratio in
the above analysis (Fig. 8), and the three cases were used to represent the upper limit of
their contribution to $NO_3$ loss. Case 1 represents the overrated contribution of $N_2O_5$ uptake
by taking $\gamma(N_2O_5) = 0.08$, which was the high value reported in high $N_2O_5$ and $ClNO_2$
plume of Shenzhen (Niu et al., 2022) and approximately seven times the parameterized
value at this site. In this case, the fraction of $NO_3$+VOCs and $N_2O_5$ uptake was significantly
elevated to account for approximately 30% of $NO_3$ loss. Case 2 shows the total $NO_3$
reactivity reached an average of $5.0 \times 10^{-3}$ $s^{-1}$ by taking $\beta$-pinene as the total monoterpene
because of the higher reaction rate constant. The weak change in the loss ratio indicates the




reactions of NO$_3$ with VOC may not be sensitive to the weights of monoterpenes, since the
contribution of monoterpenes to the NO$_3$ reactivity is not dominant. Case 3 is the synthesis
of Case 1 and Case 2 by considering higher N$_2$O$_5$ uptake coefficient and higher k(NO$_3$) to
represent the upper limit of N$_2$O$_5$ uptake and NO$_3$ reaction with VOCs to NO$_3$ loss, whose
result is slightly higher than the contribution of Case 1. Nevertheless, the quantified upper
contribution was still less than half. Thus, we conclude that most of the NO$_3$ loss was not
well accounted for even considering the uncertainties.

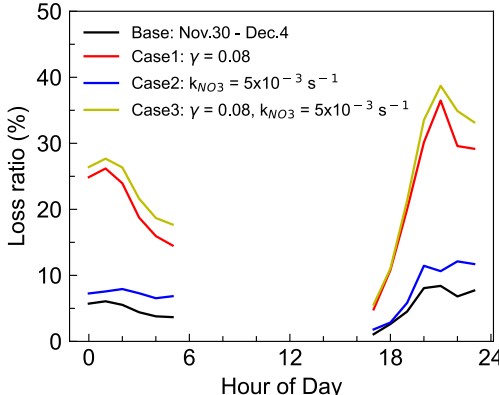


**Figure 8.** Three sensitivity tests for the contribution of VOCs and N$_2$O$_5$ uptake to the NO$_3$
loss during the nighttime of period II (November 30[th] - December 4[th]). Case 1 takes γ(N$_2$O$_5$)
= 0.08, which is the high value reported in the previous study. Case 2 takes β-pinene as the
total monoterpene with a higher reaction rate constant, and Case 3 is the synthesis of the
above two cases to represent the upper limit of the contribution.
The NO$_3$ reaction with NO was often considered to be one of the dominant loss processes
during the daytime since at nighttime NO decreased to low levels, thus not considered in
the above analysis. However, by taking NO into consideration although at low
concentration levels below the detection limit of the instrument (0.4 ppbv), the contribution
of NO to the nighttime NO$_3$ loss exceeded 100% frequently as shown in Fig. S4. Due to
the rapid reaction between NO and NO$_3$, several pptv concentrations of NO could
effectively account for most NO$_3$ loss in a relatively clean coastal environment (Crowley
et al., 2011). Nevertheless, limited by NO precise measurement, we considered the
following assessments to understand the total NO$_3$ loss processes (Fig. 9). By assuming
NO at a constant value of 40-400 pptv, more than 80% of the total NO$_3$ loss can be well
explained. Although some loss remained unidentified, these results underline that NO,
often considered to be important during daytime, was the predominant NO$_3$ loss way during
nighttime at this study site. This also suggests accurate measurement of low NO




concentrations is crucial for identifying removal pathways of nocturnal $NO_3$ oxidants and
has significant implications for nighttime atmospheric chemistry. We can infer that the
nocturnal chemical $NO_3$ reactions would be largely enhanced once without NO emission
in the open ocean after the air mass passes through this site, indicating the strong influences
of the urban outflow to the downward marine areas with respect to nighttime chemistry.

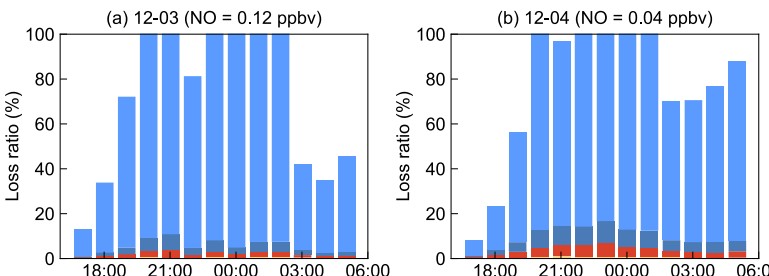

**Figure 9.** Assessments for $NO_3$ loss process by assuming NO as constant values. Blue
represents the contribution of NO and others are $N_2O_5$ uptake and VOCs.
For the absence of measured $N_2O_5$ during the CAM period, we compared the $k(NO_3)$ and
the reactivity of $N_2O_5$ uptake ($k_{het}K_{eq}NO_2$) to indirectly reflect $NO_3$ removal process.
Overall, the $NO_3$ reactivity values from VOCs and $N_2O_5$ uptake during nighttime was
relatively comparable, for 56.5% and 43.5%, respectively. This indicates that VOCs still
had a slightly larger contribution than $N_2O_5$ uptake during the CAM period, which is
consistent with the findings in southern China (Brown et al., 2016) and on the east coast of
the USA (Aldener et al., 2006).
**4.  Summary and Conclusion**
This study presents the first observation of nocturnal nitrogen oxide species, $N_2O_5$, at a
typical marine site (Da Wan Shan Island, Zhuhai) in the north of the South China Sea during
the winter of 2021. Although Da Wan Shan Island was almost free of local anthropogenic
emissions, the air pollutants from the megacities of the Pearl River Delta were transported
to this area by northerly or northeasterly winds during the measurement period.  The
maximum ratio of $N_2O_5$ was 657.27 pptv (1 min average) and the nocturnal average was
119.5 ± 128.6 pptv. The $NO_3$ production rate was comparable to that in urban areas such as
north China and the Yangtze River Delta, with an average value of 1.5 ± 0.9 ppbv h$^{-1}$ and
maximum of up to 5.84 ppbv h$^{-1}$, indicating an active nighttime chemical process in that
area.



Further analysis of $N_2O_5$ and $NO_3$ steady state lifetimes indicates that $NO_3$ had a very short
average life of $0.5 \pm 0.6$ minutes, which was to some extent comparable to that in urban
areas in summer. The combination of the high $NO_3$ production rate and short lifetime
suggests a rapid $NO_3$ loss at night. While $N_2O_5$ uptake is inefficient in relatively clean air
masses. The nighttime $k(NO_3)$ corresponded to a $NO_3$ lifetime of 4.5 minutes, indicating
that VOCs also contribute little to $NO_3$ loss. Both VOC and $N_2O_5$ uptake can only explain
less than 20% of the loss. The fast $NO_3$ loss rate also indicated the less aged air mass that
was influenced by surface-level emissions. We infer that the local weak NO emission may
significantly change the near-surface chemical pattern of $NO_3$ chemistry, which may result
in a huge difference between the observed results on the island and those on the sea surface.
We suggested that future field measurements should be made on sea surfaces away from
islands, such as ship cruise observation, to get a comprehensive understanding of the
nocturnal $NO_3$ chemistry in the background marine regions.

**Code/Data availability.** The datasets used in this study are available at:
https://doi.org/10.5281/zenodo.8089100 (Wang et al., 2023).
**Author contributions.** H.C.W. and Y. J.T. designed the study. J.W. and H.C.W. analyzed
the data with input from H.J.H., Z.L.Z., G.Z.F., C.Z.S., Z.H.L., J.Z., S.J.F.. H.C.W., L.M.
Y. J.T., Z.H.L., and J.Z. organized this field campaign and provided the field measurement
dataset. J.W., H.C.W., and Y.J.T. wrote the paper. All authors commented on and edited the
manuscript.
**Acknowledgments**
This work was supported by the National Natural Science Foundation of China (Nos.
42175111), the Guangdong Major Project of Basic and Applied Basic Research (No.
2020B0301030004), Guangdong Basic and Applied Basic Research Foundation
(2022A1515010852), and the Fundamental Research Funds for the Central Universities,
Sun Yat-sen University (23lgbj002, 23hytd002). L.M. acknowledges the Zhuhai Science
and Technology Plan Project (ZH22036201210115PWC).
**Competing interests**. The authors declare that they have no conflicts of interest.
**Appendix A Supplementary data**
Supplementary data associated with this article can be found in the online version at xxxxxx.

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
