# Peer review of "Supporting information for"

_EGUsphere, 2023_

## Author Comment (AC1)

Response to Editors and Reviewers
Manuscript ID: egusphere-2023-1401.

We appreciate the reviewers for their careful reading and constructive comments on our manuscript. As detailed below, the reviewer's comments are shown in black, our response to the comments is in blue. New or modified text is in red.
All the line numbers refer to the revised Manuscript.

Response to Referee #1:

Comments:

Wang et al., present a field measurement report about dinitrogen pentoxide ($N_2O_5$) and relevant parameters in an island site in the South China Sea, where the nighttime chemistry is less studied compared with those in urban regions in China. They showed that this site is strongly affected by the outflow of urban polluted plumes from the Pearl River Delta, China, although the local anthropogenic emission is weak and has ~50 km from the coastline. High nitrate radicals ($NO_3$) production rate, moderate $N_2O_5$ concentrations, and short $N_2O_5$ lifetime are well characterized in the outflow plumes. The budget analysis is also convincing especially in the aspect of volatile organic compounds oxidation during the nighttime.

The data presented in this report has high quality although only valid for about half a month, this data set is valuable with respect to the nighttime chemistry in the marine regions that are frequently affected by anthropogenic activities, which can be helpful to the understanding of the interactions of anthropogenic and marine air masses. The results inspire that human emission in the coastal cities may have a significant impact on the air quality in the marine regions over a large spatial scale. Overall, the paper is well written and certainly within the scope of the measurement report type in ACP. I would like to recommend minor revisions before the publication.

Thanks for the review's positive comments.

General comments:

1. As shown in Figure 2, the $N_2O_5$ data is also available in one night (11-14), but it is not presented in Figure 3 as well as Table 2, I can understand the limited data did not have representativeness of average condition for CAM, but it should be clarified in the legend of Figure, Table, and the main text.

We highly acknowledged this suggestion. We have the statement in the Line 341 as "Daytime $N_2O_5$ and $NO_3$ in the IAM period were shown as NaN due to the absence of observation." Also, we added the explanation in the note of Table 2 and in the caption of Figure 3 as follows.

Line 311: [c] Without $N_2O_5$ measurement in the daytime and limited $N_2O_5$ data during the CAM period, $N_2O_5$, $NO_3$, and their lifetimes were not valid here.

Line 350: Figure 3. Mean diurnal profiles of $N_2O_5$, $NO_3$, $P(NO_3)$, and relevant parameters in the two types of air masses. $NO_3$ was calculated from $N_2O_5$. Neither $N_2O_5$ nor $NO_3$ was shown during CAM period because of limited $N_2O_5$ measurement.

2. I strongly encourage the author to conduct more analysis about the nocturnal oxidation capacity of the different types of VOC by considering the nighttime ozone oxidation as well as the nitrate radicals.

Thanks, we added more discussion about the oxidation of VOC by $NO_3$ and $O_3$ at nocturnal time at section 3.4. The result showed that $NO_3$ accounted for 63.1% of the nocturnal VOC oxidation related to $O_3$.

Line 523. To better understand the nocturnal oxidation of VOCs, we compared the nighttime oxidation of VOCs by $NO_3$ with $O_3$. Since OH was not measured and OH is often regarded as a vital daytime oxidant (Finlayson-Pitts, 2000; Lu et al., 2010), we did not consider OH oxidation in the nighttime. Figure S4 showed the diurnal pattern of VOC loss rate by $NO_3$ and $O_3$, $NO_3$ predominantly achieves its peak oxidation rates (0.07 ppbv $h^{-1}$) during the initial half of the night, accounting for 63.1% of the total VOC oxidation on nocturnal average. Meanwhile, $O_3$ also makes a contribution to VOC oxidation, mainly owing to its relatively high nighttime concentration levels (42.9 ± 18.4 ppbv).

[Figure]

Figure S4. Diurnal profiles (mean ± standard deviation) of VOC oxidation rate by atmospheric oxidants, $NO_3$ and $O_3$. The pie chart represents the nocturnal fractions of these two oxidants to VOC oxidation.

Finlayson-Pitts, B. J., James N.: Chemistry of the upper and lower atmosphere: theory, experiments and applications, Academic Press, Calif2000.

Lu, K. D., Zhang, Y. H., Su, H., Brauers, T., Chou, C. C., Hofzumahaus, A., Liu, S. C., Kita, K., Kondo, Y., Shao, M., Wahner, A., Wang, J. L., Wang, X. S., and Zhu, T.: Oxidant $(O_3 + NO_2)$ production processes and formation regimes in Beijing, Journal of Geophysical Research-Atmospheres, 115, Artn D07303, 10.1029/2009jd012714, 2010.

3. Line 402, the concentration of phenol and cresol is below 10 ppt on average, considering the high contribution to the $NO_3$ reactivities, I suggest the author add some discussion about the instrumental detection limit of the species.

   Thanks. We added some discussion about the measurement uncertainties may be caused by the instrumental detection limit as following.

   Line 446. Considering that the measured phenol and cresol concentration is low and near the instrumental detection limit, we note this may bring some uncertainties in quantifying the contribution to the total $NO_3$ reactivity and $NO_3$ loss rate.

4. Line 515, the two cases presented in Figure 9 are 0.04 ppb and 0.12 ppb, which is not consistent with the statement of 40-400 ppt, the author should clarify it. By the

way, I don't know why the two concentrations of NO are chosen in the two cases, respectively, since the budget is still not closed in the second half of the night.

Thanks. Here we choose the fixed NO concentration that can explain about 80% of the budget on average. Figure 9 illustrates two representative examples from many days. These results show that the required NO concentration varies for each day, yet consistently remains below the detection limit of the instrument. We change the caption of Figure 9 as follows and updated the figure.

[Figure]

Figure 9. Examples for the assessment of $NO_3$ loss process by assuming NO as constant values to approximately explain about 80% of the budget.

5.  The reaction rate constants of $NO_3$ with VOCs can be added as a table in the support information.

    Thanks for your comment. We have added the reaction rate constants of $NO_3$ with VOCs in the supporting material as Table S2.

**Table S2.** Reaction rate coefficients of VOC with $NO_3$ used in this study.

| | VOC | k (298K) ($10^{-12}$ cm$^3$ molecule$^{-1}$ s$^{-1}$) | A Factor ($10^{-12}$ cm$^3$ molecule$^{-1}$ s$^{-1}$) | Ea/R (K) | Ref |
|---|---|---|---|---|---|
| | Anthropogenic compound | | | | |
| 1 | Phenol | 3800 | / | / | 1 |
| 2 | Cresol | 14000 | / | / | 1 |
| 3 | Formaldehyde | 0.00056 | / | / | 1 |
| 4 | Hexanal | 0.0027 | / | / | 1 |
| 5 | i-Butane | 0.106 | / | / | 1 |
| 6 | n-Butane | 0.046 | / | / | 1 |
| 7 | Indene | 4.1 | / | / | 1 |
| 8 | Styrene | 1500 | / | / | 1 |
| 9 | Toluene | 0.07 | / | / | 1 |
| 10 | cis-2-Pentene | 581 | / | / | 1 |
| 11 | trans-2-Pentene | 647 | / | / | 1 |
| 12 | 1-Pentene | 15 | 0.39 | 0 | 2 |
| 13 | cis-2-Butene | 352 | 0.35 | 0 | 2 |
| 14 | trans-2-Butene | 390 | / | / | 1 |
| 15 | n-Pentane | 0.087 | 3.05 | 3060 | 2 |
| 16 | Acetylene | 0.21 | / | / | 3 |
| 17 | Benzene | 0.03 | / | / | 4 |
| | Biogenic compound | | | | |
| 18 | Isoprene | 700 | 3.15 | 450 | 1 |
| 19 | α-Pinene | 1190 | 1.19 | -490 | 1 |
| 20 | β-Pinene | 6160 | / | / | 1 |
| 21 | DMS | 1100 | / | / | 5 |
| 22 | Propane | 0.03 | / | / | 1 |
| 23 | Propene | 9.49 | / | / | 1 |
| 24 | 1-Butene | 14 | 3.3 | 2880 | 2 |

Ref1: (Atkinson et al., 2003)
Ref2: (Brown et al., 2011)
Ref3: IUPAC
Ref4: Estimated
Ref5: (Brown et al., 2012)

Atkinson, R. and Arey, J.: Atmospheric degradation of volatile organic compounds, Chem Rev, 103, 4605-4638, 10.1021/cr0206420, 2003.
Brown, S. S. and Stutz, J.: Nighttime radical observations and chemistry, Chem Soc Rev, 41, 6405-6447, Doi 10.1039/C2cs35181a, 2012.
Brown, S. S., Dube, W. P., Peischl, J., Ryerson, T. B., Atlas, E., Warneke, C., de Gouw, J.

A., Hekkert, S. T., Brock, C. A., Flocke, F., Trainer, M., Parrish, D. D., Feshenfeld, F. C., and Ravishankara, A. R.: Budgets for nocturnal VOC oxidation by nitrate radicals aloft during the 2006 Texas Air Quality Study, Journal of Geophysical Research-Atmospheres, 116, Artn D24305 10.1029/2011jd016544, 2011.

Technical comments:

1.  Figure 6a, the standard deviation should be added to the bar plot.

    Thanks, we have revised the boxplot of k(NO₃) for both IAM and CAM period in Figure 6a to better demonstrate the statistical distribution as follows.

[Figure]

**Figure 6.** (a) Distributions of k(NO₃) from AVOC and BVOC for both IAM and CAM period. The error bar indicates the standard deviation. (b) The relative contribution of VOC categories to the k(NO₃).

2.  Figure 5c, the font size of the percentages is too small, it can be further improved.

    Thanks for your suggestion and we updated Figure 5 as follows. Due to the limited space, I have tried to make the font size of percentage larger to facilitate the readability.

[Figure]

3. Line 138, 3000 change to 3, 000.

Thanks, and we revised it accordingly.

---

## Author Comment (AC2)

Response to Editors and Reviewers
Manuscript ID: egusphere-2023-1401.

We appreciate the reviewers for their careful reading and constructive comments on our manuscript. As detailed below, the reviewer's comments are shown in black, our response to the comments is in blue. New or modified text is in red.
All the line numbers refer to the revised Manuscript.

Response to Referee #2:

Comments:

Wang and coauthors describe a recent study of nighttime $N_2O_5$ and $NO_3$ chemistry at a marine site in the South China Sea that was alternately influenced by inland and coastal air. They measured $N_2O_5$, $NO_x$, $O_3$, VOCs, and meteorological parameters with a suite of instruments, and calculated the expected $NO_3$ based the $N_2O_5$-to-$NO_3$ equilibrium constant. They calculated the $NO_3$ production rate and attribute its loss processes to a combination of the reaction with NO, heterogeneous uptake of $N_2O_5$ on aerosol, and reactions with of $NO_3$ with VOCs, and find that the loss is dominated by reaction with NO, even at very low NO concentrations. They compare the concentrations and lifetimes of $NO_3$ and $N_2O_5$ with other recent urban, costal, and marine studies.

The study adds to our understanding of nighttime pollution processes, and may be suitable for publication, but there are some key details missing in the description of the instrumentation and the data analysis that need to be addressed first. I recommend major revisions.

Thanks for the review's positive comments.

**General comments:**

- The authors essentially have three categories for $NO_3$ loss: $N_2O_5$ uptake, $NO_3$ + VOCs, and $NO_3$ destruction by NO. But the $NO_3$ + NO reaction makes two $NO_2$. With an excess of $O_3$ available, the $NO_2$ molecules with react to regenerate two $NO_3$ molecules, unless it reacts instead with another oxidant such as $RO_2$. So in that sense, NO destruction could be considered a null cycle. I am concerned that the authors' statement on line 466 that "nighttime $NO_3$ chemistry may be almost negligible" isn't the right conclusion to make once you consider that NO + $NO_3$ is part of the nighttime chemical cycling, which $NO_3$ + VOCs and $N_2O_5$ uptake represent termination steps. Have the authors considered instead of

looking at production of $NO_3$, looking at some kind of "net" production that accounts for the reformation of $NO_3$ back into the budget?

In line 466 of the previous manuscript, we showed that nighttime $NO_3$ chemistry may be almost negligible just with respect to the aspect of $NO_x$ loss, not to the $NO_3$ production recycle. To avoid ambiguity, we revised the description as follows.

Line 517. This result shows that the nighttime NO3 chemistry may be almost negligible to the $NO_x$ removal compared with the day $OH + NO_2$ pathway according to previous works reported in urban regions (Wang et al., 2017b; Wang et al., 2020a).

Wang, H., Lu, K., Chen, X., Zhu, Q., Chen, Q., Guo, S., Jiang, M., Li, X., Shang, D., Tan, Z., Wu, Y., Wu, Z., Zou, Q., Zheng, Y., Zeng, L., Zhu, T., Hu, M., and Zhang, Y.: High $N_2O_5$ Concentrations Observed in Urban Beijing: Implications of a Large Nitrate Formation Pathway, Environmental Science & Technology Letters, 4, 416-420, 10.1021/acs.estlett.7b00341, 2017b.

Wang, H., Chen, X., Lu, K., Hu, R., Li, Z., Wang, H., Ma, X., Yang, X., Chen, S., Dong, H., Liu, Y., Fang, X., Zeng, L., Hu, M., and Zhang, Y.: $NO_3$ and $N_2O_5$ chemistry at a suburban site during the EXPLORE-YRD campaign in 2018, Atmospheric Environment, 224, 10.1016/j.atmosenv.2019.117180, 2020a.

In addition, we totally agree with the comment that NO titration $NO_3$ can speed up the $NO_3$ production. As the reviewer suggested, we checked the role of NO in regulating the $NO_2$ and $NO_3$ chemistry by using a box model simulation. We set up an initial input of 12.9 ppbv $NO_2$, 64.5 ppbv $O_3$, and fixed 0.1 ppbv NO, 0.003 s$^{-1}$ $k_{NO3}$ and 0.0002 s$^{-1}$ $k_{N2O5}$ in the box model (typical condition at 18:00 in IAM case), the two typical scenarios with and without $NO+NO_3$ with the same input presented an additional 23% $NO_3$ production after evolution after 12 hours, which suggested that this reaction play an important role in regulating the partitioning of total oxidants, $O_x = NO_2 + O_3$, as well as the nighttime oxidation capacity if we using $P(NO_3)$ as an indicator, while it should be note that this only existed when ozone excessed condition as the reviewer mentioned.

- The authors separate the model study into two types of air masses: inland air masses (IAM) and coastal air masses (CAM). The HYSPLIT trajectories show that the CAM air follows the southeastern coast of China, though only one example trajectory was shown. How much variability was observed in the trajectories? Did any trajectories imply an influence from the megacities on the coast? Unless the authors observed distinct tracers of purely marine air, it seems like the CAM air is also continental air, just more aged than the IAM air. But in several places

such as line 332, the CAM-influence measurements are described as coming from "clean areas". The site is also described as an "island" site (i.e. Table 2), but based on wind speeds, the site is only about 2-3 hours downwind from the major cities. So I would suggested reframing these descriptions as IAM = fresh urban emissions, and CAM = aged urban emissions.

Thanks. The HYSPLIT model was run for 48 hours backward at local times of 20:00, 24:00, and 04:00 from Nov. 9th to Dec. 5th 2021. The trajectory in Figure 1b was not one trajectory, but the clusters of IAM and CAM. The IAM cluster is composed of 73 backward tracks and CAM cluster had 14 backward tracks. No air mass coming from clean ocean was mentioned at section 2.1 in line 152 as "No air masses free of pollution from the South China Sea were observed during the measurement period." Besides, the description of "clean areas" in Line 332 (in the previous manuscript) refers to the air masses of Shenzhen were from clean areas or the sea surface. To avoid ambiguity, we revised the description as follows.

Line 364. The P($NO_3$) of CAM was consistent with the observation when the air mass over eastern Shenzhen was transported from clean area or sea surface (1.2±0.3 ppbv h$^{-1}$, Niu et al., 2022).

We agreed that the CAM air masses were the aged urban emissions from Shenzhen, Hongkong and other coastal cities as well as the land-ocean boundary. We also agreed the definition of IAM = fresh urban emissions, and CAM = aged urban emissions as they were both originate from urban regions. So, we verified our manuscript as follows.

Line 148. It confirmed that the airmass during nighttime mostly came from inland China (fresh urban emissions, IAM, 84%) and the coastal areas (aged urban emissions, CAM, 16%).

- More details are required for the methods section. For example, the CEAS technique measures $N_2O_5$ by thermally converting $N_2O_5$ to $NO_3$, but there is also some ambient $NO_3$ in the measured sample. How do you solve for [$NO_3$] in on line 216 if [$N_2O_5$] is actually [$N_2O_5 + NO_3$]? This needs to be described in more detail. Additionally, the PTR-MS measurement is not described in enough detail. How were sensitivities for each compound assessed? How were backgrounds and calibrations done? Which of the VOCs were measured by PTR-MS and which by canister and was there overlap in the species that allowed a comparison to assess instrument accuracy? Please see the specific comments below for additional questions?

Thanks for your suggestion. We added additional details about the instrument in section 2.2 of the manuscript. Indeed, the CEAS measurement primarily captures $[N_2O_5 + NO_3]$, and represents $N_2O_5$ under high NOx (or low temperature) conditions when the $NO_3$-to-$N_2O_5$ ratio is likely to be low. However, refereeing to the instrument transmission efficiency ($55 \pm 6$ % for $NO_3$ and $85 \pm 3$ % for $N_2O_5$, Wang et al., 2017), we conducted an estimation that if 55% of the ambient $NO_3$ residue in the detection cell in CEAS and transforms into $N_2O_5$ through thermodynamic equilibrium, and considering the average value of $NO_2$ (13.9 ppbv) and temperature (19.9 °C) from Table 2, the $NO_3$ to $N_2O_5$ results only contributed about 3.5% to $N_2O_5$ concentrations. Nevertheless, we have taken it into account during the $N_2O_5$ data correction. To clarify this point further, we added the explanation in Line 186 as follows.

Line 186. Here the CEAS measurement encompasses the combined concentration of ambient $[N_2O_5 + NO_3]$ and effectively represents $N_2O_5$ under high NOx (or low temperature) conditions when the $NO_3$-to-$N_2O_5$ ratio is likely to be low. Accounting for the instrument's transmission efficiency and the thermal transformation between $NO_3$ and $N_2O_5$, the contribution of $NO_3$ is sufficiently negligible in comparison to $N_2O_5$. Nevertheless, we have taken it into account during $N_2O_5$ data correction.

The details mentioned about PTR-MS are added in section 2.2 as follows.

Line 211. At the end of this campaign, background measurements and instrument calibration were conducted with high-purity nitrogen and multi-component VOC gas standards, respectively. The instrument calibration results yielded strong linear relationships ($R^2 = 0.98$) between the proton transfer reaction rate constants and the sensitivities of ten calibrants, including acetaldehyde, acetone, dimethyl sulfide, isoprene, methyl ethyl ketone, benzene, toluene, styrene, o-xylene, and trimethylbenzene. The sensitivities of the uncalibrated species were determined through the rate constants of the proton transfer reactions and their correlation coefficients with sensitivity.

- The parameterization of the $N_2O_5$ uptake coefficient is not yet well understood, with many different parameterizations out there. Numerous papers have shown that there are other dependences besides temperature and relative humidity, including aerosol composition, aerosol pH, and aerosol liquid water content. This study didn't measure aerosol composition, so it would be difficult to parameterize here, but the authors should discuss why this parameterization was selected, given that the results in Figure 8 show the results are highly sensitive to $N_2O_5$.

We agree that many different parameterizations are widely used in previous studies and we mentioned it in section 2.3 (line 248 - 253). Given that some essential parameters were not directly measured during this campaign, we employed two approaches to estimate the $N_2O_5$ uptake coefficient. The first method is the pseudo steady state method by assuming that $N_2O_5$ and $NO_3$ have achieved a steady state (Brown et al., 2009). However, as illustrated in Figure S3, the presence of negative slopes or intercepts in our results suggests that this method is not suitable for application in this study. The second is the parameterization method. There are several kinds of parameterization methods proposed by using different observed parameters. As we did not measure the aerosol compositions that often be used to estimate the $N_2O_5$ uptake coefficients, so we can only use other methods that do not need these parameters. Here a simplified parameterization is available that based on relative humidity (RH) and temperature (Hallquist et al., 2003; Kane et al., 2001; Evans and Jacob, 2005). Although this method is simple, it had an overall good performance in China (Wang et al., 2022; Tham et al., 2018a; Wang et al., 2020b). Thus, we selected the parameterization method in the follow analysis. We clarified it further as follows.

Line 253. There are several kinds of methods proposed to quantify or estimate $\gamma(N_2O_5)$ by using observed parameters. Given that some essential parameters were not directly measured during this campaign, only two approaches were employed to estimate the $N_2O_5$ uptake coefficient. The first method is the pseudo steady state method by assuming that $N_2O_5$ and $NO_3$ have achieved a steady state

(Brown et al., 2009). $\gamma(N_2O_5)$ and $k_{NO_3}$ can be determined from the slope and

intercept of linear regression of $K_{eq}[NO_2]\,\tau(N_2O_5)^{-1}$ versus $0.25cSaK_{eq}[NO_2]$ respectively as shown in Eq. 8. The second is the parameterization method. As the aerosol compositions used to estimate the $N_2O_5$ uptake coefficients were not measured, only a simplified parameterization is available that based on relative humidity (RH) and temperature (Eq. 9) (Hallquist et al., 2003; Kane et al., 2001; Evans and Jacob, 2005). Although simple, it had an overall reasonable performance in China (Wang et al., 2022; Tham et al., 2018a; Wang et al., 2020b).

$$K_{eq}[NO_2]\tau(N_2O_5)^{-1} = \frac{1}{4}cS_a\gamma(N_2O_5)K_{eq}[NO_2] + k_{NO_3} \quad (\text{Eq. 8})$$

Brown, S. S., Dube, W. P., Fuchs, H., Ryerson, T. B., Wollny, A. G., Brock, C. A., Bahreini, R., Middlebrook, A. M., Neuman, J. A., Atlas, E., Roberts, J. M., Osthoff, H. D., Trainer, M., Fehsenfeld, F. C., and Ravishankara, A. R.: Reactive uptake coefficients for $N_2O_5$ determined from aircraft measurements during the Second Texas Air Quality Study: Comparison to current model parameterizations, Journal of Geophysical

Research-Atmospheres, 114, Artn D00f10 10.1029/2008jd011679, 2009.

Hallquist, M., Stewart, D. J., Stephenson, S. K., and Cox, R. A.: Hydrolysis of $N_2O_5$ on sub-micron sulfate aerosols, Phys Chem Chem Phys, 5, 3453-3463, Doi 10.1039/B301827j, 2003.

Kane, S. M., Caloz, F., and Leu, M. T.: Heterogeneous uptake of gaseous $N_2O_5$ by $(NH_4)(2)SO_4$, $NH_4HSO_4$, and $H_2SO_4$ aerosols, J Phys Chem A, 105, 6465-6470, Doi 10.1021/Jp010490x, 2001.

Evans, M. J. and Jacob, D. J.: Impact of new laboratory studies of $N_2O_5$ hydrolysis on global model budgets of tropospheric nitrogen oxides, ozone, and OH, Geophysical Research Letters, 32, Artn L09813, Doi 10.1029/2005gl022469, 2005.

Wang, H., Yuan, B., Zheng, E., Zhang, X., Wang, J., Lu, K., Ye, C., Yang, L., Huang, S., Hu, W., Yang, S., Peng, Y., Qi, J., Wang, S., He, X., Chen, Y., Li, T., Wang, W., Huangfu, Y., Li, X., Cai, M., Wang, X., and Shao, M.: Formation and impacts of nitryl chloride in Pearl River Delta, Atmos. Chem. Phys., 22, 14837-14858, 10.5194/acp-22-14837-2022, 2022.

Tham, Y. J., Wang, Z., Li, Q. Y., Wang, W. H., Wang, X. F., Lu, K. D., Ma, N., Yan, C., Kecorius, S., Wiedensohler, A., Zhang, Y. H., and Wang, T.: Heterogeneous $N_2O_5$ uptake coefficient and production yield of $ClNO_2$ in polluted northern China: roles of aerosol water content and chemical composition, Atmospheric Chemistry and Physics, 18, 13155-13171, 10.5194/acp-18-13155-2018, 2018.

Wang, H., Chen, X., Lu, K., Hu, R., Li, Z., Wang, H., Ma, X., Yang, X., Chen, S., Dong, H., Liu, Y., Fang, X., Zeng, L., Hu, M., and Zhang, Y.: $NO_3$ and $N_2O_5$ chemistry at a suburban site during the EXPLORE-YRD campaign in 2018, Atmospheric Environment, 224, 10.1016/j.atmosenv.2019.117180, 2020.

**Specific comments:**

1. Line 124 – Change "downward of the city" to "downwind of the city"

   Thanks, and we revised it accordingly.

2. Lines 140-142 – Where on the island was the measurement site? The listed latitude and longitude correspond to a location in the ocean, likely because there are not enough digits. This is relevant because the authors state several times that there may have been some local NO emissions, perhaps from soils. But they also mention that there is some local fishing boat activity. Figure S1 shows that the highest levels of NO come from the north, but must be fairly local. Could there be a local source such as a fishing boat or generator just north of the site?

   Thanks for pointing out this mistake. We have updated a more accurate latitude and longitude in Line 134 to (21°55′57″ N, 113°43′15″ E). A map extracted from google earth below may help better understand the surrounding environment. At the north of

the field site is a harbor for local residents and the other directions are predominantly covered by forests or wilderness. Thus, it is reasonable that the highest levels of NO came from the north, but may be fairly local.

Line 134. The field campaign was conducted at Da Wan Shan Island (21°55′57″ N, 113°43′15″ E) from Nov. 9th to Dec. 16th, 2021.

[Figure]

3. Line 143 – "local air flow was consistent from the northwest to southeast". Consistent seems like the wrong word here. The wind wasn't evenly distributed between those directions. Perhaps "local air flow was consistently from either the northwest or southeast".

Thanks for the correction and we revised it accordingly.

4. Line 177 – More details are needed about how mirror reflectivity was measured. What was the effective pathlength of the optical resonator?

Thanks, the mirror reflectivity was calibrated with high purity He and $N_2$ in the current experimental setup during the field measurements. We added more details about the CEAS as follows.

Line 178: A pair of high-reflectivity (HR) mirrors (Layertec GmbH, Mellingen, Germany) with a diameter of 25.0 mm (C0.00/-0.10 mm) was used to enhance the effective optical pathlength. Mirror reflectivity (R(λ)) was calibrated with high purity He and $N_2$ in the current experimental setup during the field measurements. R(λ) was

calibrated to be 0.99997, and the effective pathlength of the optical resonator was 13.96 km.

5. Line 180 – "The loss of $N_2O_5$ in the sampling line and filter was considered in the data correction". More detail, or a reference to a previous paper, is needed here.

Thanks. We considered the filter loss and wall loss in the sampling line and detection cell according to previous work (Wang et al., 2017) and did data correction according to the transmission efficiency. We added the reference in Line 184.

Line 184. The loss of $N_2O_5$ in the sampling line and filter was considered in the data correction according to previous work (Wang et al., 2017a).

Wang, H., Chen, J., and Lu, K.: Development of a portable cavity-enhanced absorption spectrometer for the measurement of ambient $NO_3$ and $N_2O_5$: experimental setup, lab characterizations, and field applications in a polluted urban environment, Atmos Meas Tech, 10, 1465-1479, 10.5194/amt-10-1465-2017, 2017a.

6. Line 200 – "VOCs were also sampled every 2 h using 2 L canisters on the days when the hourly $O_3$ mixing ratio exceeded 70 ppbv". Why was it only sampled only those days? And if the peak $O_3$ was only reached in the afternoon, how could the full day before the peak be sampled?

Thanks. In order to further enhance our understanding of ozone pollution processes, we intensified the measurement of VOCs during ozone pollution episodes. Given that PTR-TOF-MS can only measure a subset of VOC species, we utilized canister sampling to provide a more comprehensive datasets of VOCs. These datasets enabled us to conduct a detailed chemical analysis of ozone pollution origins. We indeed collected VOCs canister after $O_3$ pollution occurred, and did not measure VOCs before the pollution. Nevertheless, as we did not use the measurement results of canister sampling in this study, it does not impact the conclusions drawn in this paper. Here we made the following modifications to the manuscript:

Line 218. Meanwhile, the VOCs were also sampled by canisters and analyzed by a gas chromatograph equipped with a mass spectrometer or flame ionization detector (GC-MS) for some ozone polluted days. For the absence of nocturnal data from canister samples, the following analysis was based on the PTR-TOF-MS measurement except the weight of α-pinene and β-pinene detected by GC-MS.

7. Line 191 – More details, or a reference, should be included to describe how the SMPS inversion was calculated to generate the particle size distributions. Additionally,

information about the peak diameter, whether it changed between CAM and IAM, would be useful information.

Thanks, we added the reference of (McMurry, et al 2000) in the manuscript at Line 199.

Line 199. Aerosol surface area density ($S_a$, $\mu m^2$ $cm^{-3}$) was calculated based on the particle numbers and geometric diameter, which was calculated through the results measured by a laboratory-assembled scanning-mobility particle sizer (SMPS) according to Mcmurry et al. (2000).

McMurry, P. H., K. S. Woo, R. Weber, D. R. Chen, and D. Y. H. Pui (2000), Size distributions of 3 – 10 nm atmospheric particles: Implications for nucleation mechanisms, Philos. Trans. R. Soc. Lond. Ser. A Math. Phys. Eng. Sci., 358(1775), 2625 – 2642.

The peak diameter were < 120 nm during the whole campaign and showed small differences between IAM and CAM period, indicating no significant difference between two air masses with respect to the aerosol diameters. We added the peak diameter information as follows.

Line 393. While in this site, measurement indicated that the peak diameter in the particle number distribution was small during the whole campaign and indicated no significant difference between two air masses with respect to the aerosol diameters (Fig. S2).

[Figure]

Figure S2. Peak diameter distribution during IAM and CAM period (5 min time resolution).

8. Figure 2 – The y-axis for $NO_3$ should clearly state that it was calculated, not measured, to prevent misunderstanding

Thanks, we revised the y-axis for $NO_3$ in Figure 2 to "NO$_3$_cal (pptv)", and added annotation in the caption as follows.

[Figure]

**Figure 2.** Time series of $N_2O_5$, $NO_3$, $P(NO_3)$, $NO$, $NO_2$, $Sa$, temperature, and relative humidity in 1-hour average. $NO_3$ was calculated by measured $N_2O_5$ according to the thermal equilibrium. The light gray shadow indicates the nighttime period. The ribbon at the top separates the air masses into two categories, yellow for inland air masses (IAM) and blue for coastal air masses (CAM).

9. Line 259 – 261 – It is not clear what is meant when the authors say the $O_3$ "hourly maximum level" was exceeded "for 6 days out of 37 days of measurements". Was $O_3$ above the standard for all 24 hours of those days? Or at least 1 hour?

Here the criteria is once the maximum hourly average $O_3$ exceeds the standard, we mark this day as an $O_3$ polluted day. Therefore, it should be at least 1 hour as you mentioned. We revised the statement as follows.

Line 288. The average and maximum concentrations of ozone were $48.2 \pm 18.2$ ppbv and 120.1 ppbv, respectively. Once the maximum hourly average $O_3$ exceeded the Chinese national air quality standard (200 $\mu g\ m^{-3}$, equivalent to 93 ppbv), we marked this day as an $O_3$ pollution day. There are 6 $O_3$ polluted days out of 37 days during the campaign and all occurred during IAM periods.

10. Line 273 – What is the LOD for $NO$? That wasn't stated in the description of the instrument.

Thanks, the LOD of NO was mentioned at section 2.2 (Table 1) and we added the brief description in Line 196 as follows.

Line 196. The nitrogen oxide analyzer uses the chemiluminescence detection method to measure the original NO and converted $NO_2$, and the LOD was 0.4 ppbv for each species.

11. Line 276 – "NO is likely to come from a local source such as soil emission". Could boats, generators, or cooking emissions be responsible for this emission?

Yes, we agreed that NO comes mainly from local emissions, and the local boats, generators, or cooking may also be the source of NO. To avoid misunderstanding, we change the statement as follows:

Line 305. We infer that NO is likely originated from a local source such as soil emission, boats, cooking, and so on.

12. Line 281 – "only lasting three days". But there was lots of data missing. How many days was $N_2O_5$ actually measured? Those three days might be a significant fraction of them.

Actually, $N_2O_5$ was measured only 17 days during the whole campaign, and the $N_2O_5$ exceeded 400 pptv only at the first three days. We don't know if it also has high concentrations at the missing days. We change the statement as follows:

Line 313. $N_2O_5$ was at a moderate level on most days with a nocturnal average of 119.5 ± 128.6 pptv, with high concentrations (>400 pptv in 1-hour average) in the first three days during this campaign.

13. Line 294 and Table 2 – Does "All day" and "daily average" mean the 24-hour average or just the daytime hours? And why do these two numbers not match?

Sorry for using this ambiguous statement. The "All day" and "daily average" both mean the 24-hour average. While the "Day time" in Figure 5 and 6 means just the daytime hours (6:00-18:00). For gas species (ie. $O_3$, $NO_x$) and meteorological parameters, the "all day" average includes the daytime and nighttime concentrations while the "nighttime" average only includes nighttime concentrations. Their concentrations vary between daytime and nighttime so that these numbers are not matched. We have added the description in Table 2 - NOTE to demonstrate it.

[b] "All day" means the 24-hour average and the "Nighttime" means the time between 18:00-06:00 local time.

14. Line 298 – Where does 1.4 ppbv / hour number come from?

The $1.4 \pm 0.7$ ppbv h$^{-1}$ was the average value of nocturnal P(NO$_3$) during this campaign. We verified the statements as follows.

Line 331. The nocturnal average P(NO$_3$) during this campaign was $1.4 \pm 0.7$ ppbv h$^{-1}$, which is higher than the average value in the warm season of China with $1.07 \pm 0.38$ ppbv h$^{-1}$ (Wang et al., 2023).

15. Line 300 – It should be easy to calculate how much the rate constant will change as the temperature increases. The authors should state what percentage increase they would expect given their higher temperatures.

Thanks. We have added an example for the change of reaction rate constant with temperature as follows:

Line 335: Besides the high NO$_2$ and O$_3$, the high reaction rate constant for NO$_2$ and O$_3$ due to the high temperature at this site is a potential explanation for the high P(NO$_3$) values observed in this study (i.e., at same NO$_2$ and O$_3$ level, if temperature increased from 10 °C to 20 °C, the reaction rate constant would increase from $2.27 \times 10^{-17}$ to $3.05 \times 10^{-17}$, which means the P(NO$_3$) would be 1.34 times faster.)

16. Line 306 – If there is no data at a given time, it should be given as a blank or a null, not a 0. A zero implies that the data was measured at 0.

Thanks, we corrected it accordingly.

17. Line 314 – Can the authors estimate how much nocturnal NO$_2$ occurred? After all, they say that NO + NO$_3$ → 2NO$_2$ is the dominant pathway for NO$_3$.

Considering the anti-correlation of O$_3$ and NO$_2$ did not have statistical significance in Figure S2, we deleted this description in the revised version. The nocturnal NO$_2$ production by the recycle of NO$_3$+NO can reach more than 10 ppbv as mentioned in the response for the first comment.

18. Figure 3 – These data should have error bars to show the variability in the average diurnal cycle. Also see the note about line 306 regarding non-data being labeled as "zero"

Thanks. We added the error bar in Figure 3 with the "NaN" data removed.

[Figure]

19. Line 324 – VOC concentrations are described here as being "higher", but no VOC data are shown in either the main text or the supplement.

Thanks for the suggestion. We added the mean VOC concentrations of different air masses in the supplement Table S1and added a statement in Line 358. The detail analysis of the VOC speciation and their relation with $O_3$ formation are subjected to the work of other manuscripts, which are in preparation now.

Line 358. In addition, the higher $NO_x$ and VOC concentrations (as shown in Table S1) in the IAM period facilitated $O_3$ formation.

Table S1. The concentrations of nine most abundant VOCs (pptv) in different air masses.

| Species | Period | |
| --- | --- | --- |
| | IAM | CAM |
| Propene | 143±90 | 92±72 |
| Butene | 185±130 | 112±101 |
| Pentenes | 14±8 | 10±6 |
| Styrene | 11±20 | 5±4 |
| DMS | 17±6 | 18±5 |
| Isoprene | 38±20 | 29±20 |
| Monoterpene | 6±6 | 4±3 |
| Phenol | 7±3 | 6±2 |
| Cresol | 5±3 | 3±2 |

20. Line 327 – The way this sentence is phrased makes it sounds like 155 pptv was the peak for both $N_2O_5$ and $NO_3$. Rephrase this for clarity.

Thanks. We updated the peak values of $NO_3$ as follows:

Line 361. With the elevated precursor concentrations ($NO_2$ and $O_3$) in the IAM period, $N_2O_5$ and $NO_3$ accumulated rapidly after sunset, reaching their peak values (492.1 pptv and 49.6 pptv for each).

21. Figure 4. Please include the CAM/IAM bar at the top of this graph, like in Figure 2, to help the reader see which data came from which period.

Thanks, we updated Figure 4 as follows:

[Figure]

22. Line 359 – Why is 600 um$^2$/cm$^3$ considered a threshold for cleanliness? Is that a value from literature? If so, it should be referenced.

Yes, it was cited from (Wang et al., 2017b). We added the reference.

Wang, H., Lu, K., Chen, X., Zhu, Q., Chen, Q., Guo, S., Jiang, M., Li, X., Shang, D., Tan, Z., Wu, Y., Wu, Z., Zou, Q., Zheng, Y., Zeng, L., Zhu, T., Hu, M., and Zhang, Y.: High $N_2O_5$ Concentrations Observed in Urban Beijing: Implications of a Large Nitrate Formation Pathway, Environmental Science & Technology Letters, 4, 416-420, 10.1021/acs.estlett.7b00341, 2017b.

23. Line 372 – How was distinction between anthropogenic and biogenic VOCs made? Is there overlap between these two categories? A table would be helpful here, that also contains the rate constants for each VOC that is part of the k(NO₃) calculation, as well as references for each rate constant.

Thanks. The categorization of VOCs was based on the work of Gu et al., 2021. They categorized VOCs according to anthropogenic and biogenic emission inventories, respectively. The majority of compounds in the anthropogenic VOC (AVOC) emission inventory differ from those in the biogenic VOC (BVOC) emission inventory due to the distinct emission sources. Although there are some overlaps, such as hexanal, isoprene, we believe there is little overlap between these two classification systems due to the relatively limited emission sources on the island. We added Table S2 to show their categories and their reaction rate constants with $NO_3$ and cited the reference in Line 411.

Line 411. The $NO_3$ reactivity (k($NO_3$)) towards VOCs was calculated by Eq. 4, towards which were categorized into anthropogenic VOC and biogenic VOC (Gu et al., 2021).

Gu, S., Guenther, A., and Faiola, C.: Effects of Anthropogenic and Biogenic Volatile Organic Compounds on Los Angeles Air Quality, Environ Sci Technol, 55, 12191-12201, 10.1021/acs.est.1c01481, 2021.

**Table S2.** Reaction rate coefficients of VOC with $NO_3$ used in this study.

| | VOC | k (298K) ($10^{-12}$ cm$^3$ molecule$^{-1}$ s$^{-1}$) | A Factor ($10^{-12}$ cm$^3$ molecule$^{-1}$ s$^{-1}$) | Ea/R (K) | Ref |
|---|---|---|---|---|---|
| | Anthropogenic compound | | | | |
| 1 | Phenol | 3800 | / | / | 1 |
| 2 | Cresol | 14000 | / | / | 1 |
| 3 | Formaldehyde | 0.00056 | / | / | 1 |
| 4 | Hexanal | 0.0027 | / | / | 1 |
| 5 | i-Butane | 0.106 | / | / | 1 |
| 6 | n-Butane | 0.046 | / | / | 1 |
| 7 | Indene | 4.1 | / | / | 1 |
| 8 | Styrene | 1500 | / | / | 1 |
| 9 | Toluene | 0.07 | / | / | 1 |
| 10 | cis-2-Pentene | 581 | / | / | 1 |
| 11 | trans-2-Pentene | 647 | / | / | 1 |
| 12 | 1-Pentene | 15 | 0.39 | 0 | 2 |
| 13 | cis-2-Butene | 352 | 0.35 | 0 | 2 |
| 14 | trans-2-Butene | 390 | / | / | 1 |

| 15 | n-Pentane | 0.087 | 3.05 | 3060 | 2 |
| 16 | Acetylene | 0.21 | / | / | 3 |
| 17 | Benzene | 0.03 | / | / | 4 |
| | Biogenic compound | | | | |
| 18 | Isoprene | 700 | 3.15 | 450 | 1 |
| 19 | α-Pinene | 1190 | 1.19 | -490 | 1 |
| 20 | β-Pinene | 6160 | / | / | 1 |
| 21 | DMS | 1100 | / | / | 5 |
| 22 | Propane | 0.03 | / | / | 1 |
| 23 | Propene | 9.49 | / | / | 1 |
| 24 | 1-Butene | 14 | 3.3 | 2880 | 2 |

Ref1: (Atkinson et al., 2003)
Ref2: (Brown et al., 2011)
Ref3: IUPAC
Ref4: Estimated
Ref5: (Brown et al., 2012)

Atkinson, R. and Arey, J.: Atmospheric degradation of volatile organic compounds, Chem Rev, 103, 4605-4638, 10.1021/cr0206420, 2003.
Brown, S. S. and Stutz, J.: Nighttime radical observations and chemistry, Chem Soc Rev, 41, 6405-6447, Doi 10.1039/C2cs35181a, 2012.
Brown, S. S., Dube, W. P., Peischl, J., Ryerson, T. B., Atlas, E., Warneke, C., de Gouw, J. A., Hekkert, S. T., Brock, C. A., Flocke, F., Trainer, M., Parrish, D. D., Feshenfeld, F. C., and Ravishankara, A. R.: Budgets for nocturnal VOC oxidation by nitrate radicals aloft during the 2006 Texas Air Quality Study, Journal of Geophysical Research-Atmospheres, 116, Artn D24305 10.1029/2011jd016544, 2011.

24. Line 377 – This is the first time that outflow from Hong Kong and Shenzen in CAM-influenced air is mentioned, and should be mentioned earlier.

   Thanks, we added the statement in Line 150 where the IAM-CAM were firstly mentioned.

   Line 150. IAM features the outflow from inland China, such as Guangzhou and Changsha, while CAM features the outflow of coastal cities like Hong Kong and Shenzhen.

25. Figure 5 – See comment about Figure 4 regarding putting an IAM/CAM bar on top to help guide the eye.

   Thanks, we updated Figure 5 as follows:

[Figure]

26. Line 403 – "significantly higher" requires some quantification, especially because just before that you said "despite their lower concentrations". Are VOCs here lower or higher than expected given the location?

Thanks, although the phenol and cresol have an average value of $7 \pm 3$ pptv and $4 \pm 3$ pptv respectively, they played dominant roles in $NO_3$ reactivity. The lower concentrations of phenol and cresol were compared with other VOCs in this campaign, as most of them were tens of pptv.

While the "significantly higher" is compared with other reports, such as 14 pptv in the Strasbourg area, France (Delhomme et al., 2010) and 16 pptv in Great Dun Fell, UK (Lüttke et al., 1997). We added the specific citations in Line 450.

Line 450. These substances are mainly secondary species from aromatic compounds and higher concentrations have also been observed, such as in the Strasbourg area, France (14 pptv, Delhomme et al., 2010) and in Great Dun Fell, UK (16 pptv, Lüttke et al., 1997).

Delhomme, O., Morville, S., and Millet, M.: Seasonal and diurnal variations of atmospheric concentrations of phenols and nitrophenols measured in the Strasbourg area, France, Atmospheric Pollution Research, 1, 16-22, 10.5094/apr.2010.003, 2010.

Lüttke, J., Scheer, V., Levsen, K., Wünsch, G., Cape, J. N., Hargreaves, K. J., Storeton-West, R. L., Acker, K., Wieprecht, W., and Jones, B.: Occurrence and formation of nitrated phenols in and out of cloud, Atmospheric Environment, 31, 2637-2648, 1997.

27. Line 420 – Error bars on the calculated k(NO₃) are required here to quantify whether this is really "significantly" different between the IAM and CAM periods.

We added the standard deviation of k(NO₃) in Line 463 as follows. Moreover, we revised Figure 6a with error bars to better demonstrate the statistical distribution.

Line 463. As shown in Fig. 6a, $k(NO_3)$ differed significantly between the inland and coastal air masses, with $5.2 \pm 3.1 \times 10^{-3}$ s$^{-1}$ and $3.7 \pm 1.9 \times 10^{-3}$ s$^{-1}$ on average in IAM and CAM periods, respectively. Of which anthropogenic VOC-$k(NO_3)$ in IAM ($3.5 \pm 2.3 \times 10^{-3}$ s$^{-1}$) was higher than in CAM ($2.3 \pm 1.4 \times 10^{-3}$ s$^{-1}$) and dominant in both air masses, while biogenic VOC-$k(NO_3)$ was comparable ($1.7 \pm 0.9 \times 10^{-3}$ s$^{-1}$ and $1.4 \pm 0.6 \times 10^{-3}$ s$^{-1}$ for IAM and CAM, respectively).

[Figure]

28. Line 453 – Please define the "loss ratio". Is it (loss by process X) / (loss by all processes)?

Yes, the loss ratio, LR(NO₃), is defined as (loss rate by process X) / (total loss rate). But considering that all processes are not clear here and the lifetime of NO₃ is short (<1 min), which means the NO₃ production rate equal to the total loss rate, we used the NO₃ production rate to represent the total NO₃ loss rate. We added the definition of loss ratio as follows;

Line 498. The $LR$(NO₃) is defined as the sum of loss rate by process X (VOC or N₂O₅ uptake) to the total NO₃ loss rate, here the total NO₃ loss rate is represented by P(NO₃) since we cannot quantify the total NO₃ loss rate due to the NO concentration below the limit of instrument detection.

29. Line 453 – This phrasing is unclear. Are the authors saying that we can assume that total loss = total production because there is very little $NO_3$? Is this a valid assumption given the fact that $NO_3$ is measured above 0?

Yes, here we assume the $NO_3$ budget is overall in balance and use the $NO_3$ total production rate to represent the total $NO_3$ loss rate. According to previous studies, the steady state may be not so easy to achieve due to the NO emission interferences, while the chemical equilibrium, namely the production equals the loss rate in a short time scale, can be reached except during the beginning of sunset. To clarify this expression, we revised Line 497 as follows.

Line 497. To assess the contribution of various loss processes to the total $NO_3$ removal, we calculated their loss rate and the loss ratio, $LR(NO_3)$. The $LR(NO_3)$ is defined as the sum of the loss rate by process X (VOC or $N_2O_5$ uptake) to the total $NO_3$ loss rate, here the total $NO_3$ loss rate is represented by $P(NO_3)$ since we cannot quantify the total $NO_3$ loss rate due to the NO concentration below the limit of instrument detection.

30. Figure 9 – There are two blues in this figure. And it isn't clear what the second blue and the red traces represent from the figure caption. Please include a legend.

Thanks, we revised Figure 9 as follows:

[Figure]

1. Figure S1 – Why does this wind rose look different than the one in Figure 1?

Thanks, in Figure 1, the wind rose depicts wind direction with wind speed during the whole campaign. While the Figure S1 shows the NO level with wind direction at nocturnal time. We revised the caption of Figure S1 as follows to clarify their differences.

Figure S1. The wind rose plot for nocturnal NO concentrations (ppbv) and wind direction.

2. Figure S4 – Was this loss ratio calculated assuming NO was between 40 and 400 ppt? Or with the measured values? If the latter, it would be helpful to include the calculation with both values as well, to show the range of expected loss ratios.

The loss ratio in Figure S5 was calculated with measured NO. Now we updated Figure S5 as follows to include the loss ratio of assuming NO values with 0.04 ppbv, 0.12 ppbv, and 0.4 ppbv. The contribution of NO to loss ratio varied significantly in different days. For example, to reach a 100% loss ratio, NO concentration should be higher than 0.4 ppbv on Nov. 27[th], ~ 0.12 ppbv on Dec. 3[rd], and only ~0.04 ppbv on Dec. 4[th]. The result is also shown in Figure 9.

[Figure]

Figure S5. (a) Nighttime NO mixing ratio with the gray dashed line denoting the detection limit of the instrument (0.4 ppbv). (b) The fraction ratio of NO to $NO_3$ loss, with the black dashed line representing a maximum ratio of 100%.